# Liposome-Derived Nanosystems for the Treatment of Behavioral and Neurodegenerative Diseases: The Promise of Niosomes, Transfersomes, and Ethosomes for Increased Brain Drug Bioavailability

**DOI:** 10.3390/ph16101424

**Published:** 2023-10-08

**Authors:** Patrícia C. Pires, Ana Cláudia Paiva-Santos, Francisco Veiga

**Affiliations:** 1Faculty of Pharmacy, Faculty of Pharmacy of the University of Coimbra, Azinhaga de Santa Comba, 3000-548 Coimbra, Portugal; fveiga@ci.uc.pt; 2REQUIMTE/LAQV, Group of Pharmaceutical Technology, Faculty of Pharmacy, University of Coimbra, 3000-548 Coimbra, Portugal; 3Health Sciences Research Centre (CICS-UBI), University of Beira Interior, Av. Infante D. Henrique, 6200-506 Covilhã, Portugal

**Keywords:** Alzheimer’s disease, anxiety, brain bioavailability, depression, ethosomes, intranasal, niosomes, Parkinson’s, schizophrenia, transfersomes

## Abstract

Psychiatric and neurodegenerative disorders are amongst the most prevalent and debilitating diseases, but current treatments either have low success rates, greatly due to the low permeability of the blood–brain barrier, and/or are connected to severe side effects. Hence, new strategies are extremely important, and here is where liposome-derived nanosystems come in. Niosomes, transfersomes, and ethosomes are nanometric vesicular structures that allow drug encapsulation, protecting them from degradation, and increasing their solubility, permeability, brain targeting, and bioavailability. This review highlighted the great potential of these nanosystems for the treatment of Alzheimer’s disease, Parkinson’s disease, schizophrenia, bipolar disorder, anxiety, and depression. Studies regarding the encapsulation of synthetic and natural-derived molecules in these systems, for intravenous, oral, transdermal, or intranasal administration, have led to an increased brain bioavailability when compared to conventional pharmaceutical forms. Moreover, the developed formulations proved to have neuroprotective, anti-inflammatory, and antioxidant effects, including brain neurotransmitter level restoration and brain oxidative status improvement, and improved locomotor activity or enhancement of recognition and working memories in animal models. Hence, albeit being relatively new technologies, niosomes, transfersomes, and ethosomes have already proven to increase the brain bioavailability of psychoactive drugs, leading to increased effectiveness and decreased side effects, showing promise as future therapeutics.

## 1. High Prevalence Brain Disorders: Current Treatments and Their Limitations

Brain disorders represent a significant global burden, being estimated to affect over 1 billion people worldwide, being major causes of disability, and even increasing mortality, especially in middle- and high-income countries [1,2]. These illnesses contribute to the global disease burden quite significantly, and more than all other non-communicable diseases, such as cardiovascular diseases, or even cancer [3,4,5]. As global awareness and access to information expand, so do the number of diagnoses, but the increase in average human life expectancy also has an important impact on the estimated incidence of these diseases, since brain functions deteriorate as this organ ages [3,5]. Although psychiatric and neurodegenerative disorders are usually classified separately, they often exist simultaneously, which not only contributes to polymedication and, consequently, decreased patient treatment compliance, but also results in a more difficult diagnosis, since some symptoms tend to overlap [1,4,5]. Some of the most prevalent and disabling diseases with a brain etiology are schizophrenia, bipolar disorder, depression, anxiety disorders, Alzheimer’s disease, and Parkinson’s disease (Figure 1).

Schizophrenia is a psychiatric disorder that is usually developed in early life, due to abnormal brain development, including structural and functional changes in the regions of the medial temporal and prefrontal lobes, which are brain regions involved in declarative and working memory regulation [6,7,8]. These alterations have led to the disease being characterized by different behavioral and cognitive symptoms, which are classified as either positive (related to experiences that are “an addition to reality”), such as delusions, hallucinations, disorganization, and erratic thought patterns, or negative (related with loss of certain abilities), including slow movements, general lack of motivation, and withdrawment [7,9,10]. Patients tend to lose their capability to perform simple daily tasks or be in socially demanding situations, which at many times leads to both occupational and social decline [10,11]. Schizophrenia has a strong genetic component, being passed on generationally, and its pathophysiology is generally connected to how the brain processes information, including changes in specific brain cell populations and the communication between them, and having been associated with neurochemical imbalances in the levels of dopamine, γ-aminobutyric acid (GABA), glutamate, and serotonin [6,9,10]. Current pharmacological treatments include drugs that act on the normalization of these neurotransmitters’ levels, including dopamine D2 receptor antagonists (first-generation or typical antipsychotics, such as chlorpromazine, loxapine, and haloperidol), serotonin 5-HT2_A_ and dopamine D2 antagonists (second-generation or atypical antipsychotics, such as clozapine, quetiapine, and risperidone), and dopamine D2 partial agonists (third-generation antipsychotics, such as aripiprazole, brexpiprazole, and cariprazine), but these drugs are generally only able to treat the psychotic symptoms to some extent, and do not have a substantial impact on the cognitive, functioning, and social aspects [10,11,12]. Furthermore, although non-pharmacological adjuvant treatments, such as psychotherapy (individual or in groups, including narrative, mindfulness, or meta-cognitive training), or stimulation to medication adherence, can help, there is still much scope for improvement in the treatment of schizophrenia [6,13].

Similarly, bipolar disorders are also one of the leading disability causes among young people, being closely related to schizophrenia in the way that they also lead to functional and cognitive impairments [14,15,16]. Early diagnoses are difficult to achieve, since some symptoms overlap with other disorders, such as mood swings (schizophrenia) or depressive episodes (depression) [15,16,17]. Bipolar disorders can either be classified as type I, being mostly characterized by the existence manic episodes (uncontrollable thoughts, restlessness or hyperactivity, obsessive compulsive behavior, exaggerated emotions, and extreme mood changes, including magnified irritability or happiness), or type II, being characterized by hypomanic episodes (a less severe form of mania, with similar symptoms as manic episodes, but in a less intense manner) associated with depressive features [14,15,18]. Gene–environment interactions are believed to explain their etiology the best, having a strong genetic component (approximately 70% heritability), but also being commonly related to traumatizing events, especially those occurring in early life (adverse environmental exposures, such as childhood maltreatment) [14,17]. Its pathogenesis is believed to be related to disturbances in monoaminergic signaling, neuronal–glial plasticity, cellular metabolic pathways, inflammatory responses, and mitochondrial function [15,16]. Pharmacological treatment often includes antipsychotic (quetiapine, lurasidone, and olanzapine), antidepressant (amitriptyline, paroxetine, and bupropion), or even some antiepileptic drugs (carbamazepine, lamotrigine, and valproic acid), although lithium continues to be one of the most effective therapies, due to its general mood-stabilizing properties, with specific antidepressant and antimanic effects due to its role in the reduction of dopamine and glutamate and increase of GABA neurotransmission [15,19,20,21]. Nevertheless, treatment efficacy is known to be reduced, and the need for a patient-tailored approach leads to patient non-compliance due to the long time it can take until a proper treatment is found [15,16,18].

On the other hand, depression has such a high incidence that it not only affects the patient’s quality of life severely, but it also has a heavy impact on society in general, due to its associated healthcare costs and lowering of work productivity [22,23]. Even though there are several subtypes (major depression, persistent depressive disorder, depression associated with bipolar disorder, postpartum depression, etc.), with different symptoms (or differences in symptom intensity or frequency), overall clinically depressed individuals usually experience loss of interest and/or anhedonia, low self-esteem that comes with feelings of worthlessness, hopelessness, or guilt, mood swings or overall depressed mood, exaggerated loss of energy or general fatigue, psychomotor and sleep alterations, appetite and/or weight alterations, and cognitive difficulties, such as trouble thinking or concentrating, or general indecisiveness [24,25,26]. Patients with major depression characteristics also tend to have suicidal thoughts, which consequently is linked to an elevated degree of mortality [22,24,25]. Although psychotherapies are often sought before any other treatment, most patients end up needing pharmacological options. Since the disease’s pathophysiology is related to monoamine deficiency (reduction in serotonin, dopamine, and noradrenaline levels), most antidepressant drugs have mechanisms related to this, with options including tricyclic antidepressants (which block serotonin and noradrenaline reuptake, such as imipramine, amitriptyline, and nortriptyline), selective serotonin reuptake inhibitors (such as fluoxetine, paroxetine, and escitalopram), serotonin and noradrenaline reuptake inhibitors (such as venlafaxine, desvenlafaxine, and duloxetine), monoamine oxidase inhibitors (such as phenelzine, isocarboxazid, and moclobemide), noradrenaline and dopamine reuptake inhibitors (such as bupropion), and serotonin receptor antagonist and reuptake inhibitors (such as trazodone) [22,24,27]. Nevertheless, the treatment should again be tailored to the patient’s needs, since a substantial interindividual variability in the therapeutic outcome tends to exist, and, once again, this need for a trial-and-error approach can lead to patient non-compliance [23,24]. Additionally, even reasonably effective treatments have their limitations, being accompanied by a high prevalence of mild-to-severe systemic side effects, such as dry mouth, indigestion, nausea, diarrhea, constipation, blurred vision, headache, abdominal pain, fatigue, heat stroke, elevated blood pressure, or sudden weight loss [22,28,29].

Anxiety disorders are another type of very common psychiatric disorders, involving a brain circuit’s dysfunction in response to perceived danger [30,31]. Relevant subtypes include generalized anxiety disorder, social anxiety disorder, phobias, panic disorder, among others [30,32,33]. These disorders often come hand-in-hand with depression, and similarly to depression, anxiety disorders lead to an overall reduction of the patient’s quality of life and productivity, being associated with high health and social costs [30,31,34]. Additionally, the comorbidity of anxiety with depression usually originates from more severe symptoms and a higher difficulty in attaining an effective treatment [30]. Although some of the main symptoms overlap with those also frequently present in depression, such as trouble sleeping, anxiety disorders are usually characterized by a chronically intense nervousness or restlessness, easy irritability, worry and fear, fatigue, trouble concentrating, dizziness, stomach discomfort, headache or pain located in other places of the body, increased muscular tension, changes in heartbeat, sweating, and/or a light feeling of “needles” on the skin’s surface [30,32,33]. Non-pharmacological treatment includes psychotherapy (cognitive behavioral therapy), and pharmacological treatment includes selective serotonin reuptake inhibitors (e.g., sertraline, escitalopram, and fluoxetine), serotonin and norepinephrine reuptake inhibitors (e.g., duloxetine and venlafaxine), or the commonly used benzodiazepines (alprazolam, diazepam, and midazolam), which act as GABA receptor agonists, especially for type A [31,32,33]. Nevertheless, once more, the need for a patient-tailored treatment is likely to leave patients with a lack of an optimum therapeutic effect for long periods of time, and side effects, such as hypotension, weight gain, sedation, impaired cognitive or motor functioning, headaches, stroke, and even the risk of induced coma or death due to overdosing, can lead to a reduced patient compliance [30,31,35]. Moreover, several factors have a substantial impact in treatment efficacy, such as comorbid disorders, prior treatments, and the subtype and severity of the disease, which all further contribute to treatment failure [31,32,34].

In contrast, Alzheimer’s disease is a neurodegenerative disease, known to be a severely impairing illness, characterized by a slow but progressive loss of brain function, including cognitive decline, with memory and thinking ability failure, which often progresses to dementia [36,37,38]. It is a major cause of disability, also being connected to increased mortality, with the person developing high levels of dependence, since aside from experiencing difficulty in remembering things, which worsens with time, patients also tend to lose their ability to communicate, as well as the awareness of their surroundings [36,38,39]. Additionally, patients may produce uncharacteristic sounds (such as groaning or moaning), have increased anxiety levels, and have mood or personality changes, which can sometimes lead to aggressive behavior [36,39,40]. Its familial form is mainly genetically inherited, manifesting due to mutations in the amyloid precursor protein and in the presenilin 1 and presenilin 2 genes, and having most incidence in middle to late life, while its sporadic form is the consequence of both genetic (apolipoprotein E gene) and environmental factors (including exposure to pollutants, such as heavy metals, pesticides, and industrial chemicals), having most incidence in late life [41,42,43,44]. On a molecular level, it is characterized by β-amyloid and Tau protein accumulation in the brain, which leads to brain cell degeneration, accompanied by substantial oxidative and inflammatory process aggravations [36,37,40]. Conventional treatments either act on cholinesterase enzyme inhibition (e.g., donepezil, rivastigmine, and galantamine) or as N-methyl d-aspartate antagonists (e.g., memantine), but are only effective for symptomatic treatment, having little-to-no effect in disease progression, and hence not effectively curing or preventing it [36,39,45]. Some more recent alternatives include Tau phosphorylation inhibitors (such as memantine or sodium selenate), and immunotherapies, namely anti-amyloid monoclonal antibodies (such as lecanemab or aducanumab), which have proven to have increased efficacy in disease treatment and progression prevention, but present several problems, such as the need for adjuvant treatment and side effects related to immunogenicity [46,47,48].

Parkinson’s disease also has a neurodegenerative nature, with patients usually experiencing severe and gradually intensified movement impairments [49,50,51]. Common symptoms include tremors, postural instability, akinesia, bradykinesia, and/or rigidity, which will make the patient progressively less capable of performing even the simplest daily tasks on their own [49,51,52]. Nevertheless, non-motor symptomatology also plays a clinically significant role, including drowsiness and sleep disorders, constipation, impaired olfaction, or even depression and/or psychosis [50,52,53]. With genetics also playing an important role, and having the most incidence in the elderly, its pathophysiology mostly relates to neuronal loss, more specifically of dopaminergic neurons, in certain brain areas, such as the substantia nigra [50,51,52]. It is also characterized by the development of Lewy bodies, containing an accumulation of the protein α-synuclein [50,52,53]. The most commonly used pharmacological therapy is levodopa, which is converted into dopamine via enzymatic degradation after undergoing uptake by neuronal cells in the brain [49,51,53]. Dopamine agonists are also used, with ergot dopamine agonists mainly acting on D2, D3, and D4 receptors (e.g., cabergoline, bromocriptine, and pergolide), and with non-ergot dopamine agonists acting more selectively on D2 or D3 receptors (e.g., ropinirole and pramipexole), with anticholinergics (e.g., benzhexol, orphenadrine, and benztropine), catechol-O-methyltransferase inhibitors (e.g., entacapone, tolcapone, or opicapone), monoamine oxidase-B inhibitors (e.g., selegiline and rasagiline), and amantadine (N-methyl-D-aspartate-glutamate and cholinergic muscarinic receptor inhibitors) also being commonly prescribed [54,55,56]. Non-pharmacological treatments, such as physical exercise and physiotherapy (involving endurance, balance, strength, and coordination), along with cognitive training (memory and logical thinking), are also required in order to slow down the disease’s progression [57,58]. Nevertheless, again, these treatments can only be effective in treating the symptoms associated with Parkinson’s disease, and therapies that can slow down its progression or even cure it are still lacking [49,51,52].

Despite having some different characteristics, all these disorders tend to have a chronic nature, lasting for many years or even a lifetime, and hence have a very significant impact not only on the individuals themselves, but also in the society in which they are inserted. Additionally, as mentioned, aside from a lack of therapeutic efficacy and the need for repeated dosing regimens, and with many existing treatments only being effective for the disease’s symptoms, hence not being a “true cure”, most medications lead to a wide variety of systemic side effects, some of which are quite serious and disabling, with a highlight on sedation (benzodiazepines), weight gain that can culminate into obesity and associated problems (antidepressants and antipsychotics), neuroleptic malignant syndrome (antipsychotics and antiparkinsonian agents), cerebrovascular accidents (antidepressants and antipsychotics), induced parkinsonism (anti-Alzheimer agents), and even suicidal ideation (antidepressants and anxiolytics) [59,60,61,62,63], and, of course, taking more than one drug simultaneously will increase the propensity, intensity, and/or frequency of such side effects, which unfortunately is not a very unlikely scenario, since psychiatric diseases tend to coexist, either due to similar pathophysiology and/or causes, medication side effects, or even with one psychiatric illness leading to another (as is the case of depressive and anxiety disorders being commonly diagnosed together) [59,60,61,62].

Furthermore, aside from efficacy and safety issues, most drug molecules also have characteristics that will hinder their bioavailability, such as high first-pass metabolism, low permeation through the biological barriers (especially the blood–brain barrier (BBB)), high plasma protein binding, considerable P-glycoprotein efflux, and low water solubility, which will make it difficult to both formulate them at high strength and make them reach the intended therapeutic site at concentrations required for the therapeutic effect to occur (in this case, the brain) [64,65,66,67,68]. Nevertheless, although these problems are not easily overcome using conventional formulations, nanotechnology can be the answer.

## 2. Nanosystems as Non-Conventional Forms of Treatment for Increased Efficacy and Safety

Nanotechnology is the science that manipulates matter at the nanometric scale, and over the years it has proved to be advantageous when applied to a variety of fields, such as engineering, food, cosmetic science, and medicine, both in diagnostics and therapeutics [69,70,71,72]. In pharmaceutical nanotechnology, the development of nanosystems (structures with sizes between 1 nm and 1000 nm) for drug molecule encapsulation has proven to be beneficial in the treatment of a wide number of diseases, including cardiovascular, oncological, and neurological diseases, among many others [69,73,74,75,76].

Although nanosystems can be divided into categories according to their composition, amongst other specific characteristics, in general, all of them offer a means to protect drug molecules against chemical or metabolic degradation, due to encapsulation within structures that provide a barrier to the outside environment [77,78,79]. Additionally, they can also increase drug permeation through biological barriers, and when functionalized with specific moieties, which are molecules that will be inserted onto the surface of the nanoparticles and that will bind to specific cell surface receptors on target tissues, such as proteins and peptides (lectin, transferrin, lactoferrin, apolipoprotein E, etc.), or smaller molecules (polyethylene glycol, folic acid, galactose, mannose, etc.), will provide active drug transport into these target cells and tissues, and, hence, drug targeting, thus allowing an increased bioavailability in these tissues and decreased systemic drug distribution, leading to potentially safer and more effective therapies [77,80].

Liposomes are a specific type of spherical vesicular nanosystem, characterized by having one or more phospholipid bilayers surrounding an aqueous core (Figure 2). This structure gives this type of nanosystem the important advantage of being capable of encapsulating both hydrophobic and hydrophilic molecules in the membrane (hydrophobic nature) or in the core (aqueous nature), respectively, either separately or simultaneously [81,82,83]. Given this very advantageous versatility, and with the added advantages of being biocompatible and biodegradable, having controlled drug release, and leading to an overall increased drug absorption and, consequently, bioavailability, liposomes have received a substantial amount of attention from both academic- and industrial-based research [81,84,85]. Additionally, due to favorable results in targeting and accumulation in tumor tissues, a few liposomal formulations have already reached the pharmaceutical market for cancer treatment [81,83,84]. Nevertheless, these systems have been reported to have low stability, and premature drug leakage from the vesicles has been known to occur, which could lead to an initial burst release after administration or drug loss during formulation storage, which is a substantial disadvantage [81,82,85]. Hence, to tackle these issues, liposome modifications were developed, giving rise to new and improved vesicles for drug delivery—ethosomes, transfersomes, and niosomes (Figure 2).

Ethosomes, just like the name implies, have ethanol in their composition at high concentrations (from 20% to 50%), as well as phospholipids (such as cholesterol, phosphatidylcholine, and phosphatidylethanolamine) [84,86,87]. Common preparation methods include the cold method, the hot method, the classic mechanical dispersion method, and the transmembrane pH gradient method [86,87]. Nevertheless, despite having ethanol in their composition, and just like liposomes, ethosomes have been reported to be biocompatible, biodegradable, and generally non-toxic [84,88,89]. Additionally, aside from having higher stability (due to electrostatic repulsion and steric stabilization), higher entrapment efficiency, and a lower particle size than liposomes, and a characteristically negative zeta potential (since ethanol acts as a negative charge provider), ethosomes have also proven to lead to improved drug permeation, since ethanol is a known permeation enhancer, resulting in a transient disruption in the biological membranes’ barrier properties, due to enhancing cell membrane fluidity and reducing their multi-layered lipids density, hence promoting drug absorption [86,87,88]. Furthermore, these vesicles are soft, highly fluid, flexible, and elastic, which gives them a deformability capacity, that not only further increases drug permeation, but also allows for a higher tissue drug deposition, prolonging the desired therapeutic effect [87,88,89]. Due to these properties, these vesicles have been mostly applied for transdermal drug delivery, since they are able to transiently disrupt the stratum corneum, by dissolving and extracting the intercellular lipids that are part of its composition, but these characteristics might be useful for the disruption of other biological barriers as well, such as the BBB [90,91].

On the other hand, transfersomes (sometimes also called “transferosomes”) are likewise bilayered vesicles, similar to ethosomes in the way that they are altered liposomes, but instead of ethanol they have the addition of an edge activator, usually surfactants, such as Tweens, Spans, or sodium deoxycholate, to the lipid bilayer, usually made of phosphatidylcholine or lecithin [92,93,94]. The addition of these softening compounds to the bilayer will allow these nanosystems to have ultra-flexibility, due to elasticity and deformability capacity, which will make them able of deforming and squeezing through and across biological membranes, including through pores substantially smaller than their own size, such as intercellular gaps, while remaining intact, hence increasing drug permeation [84,92,95]. These edge activators also have the capability of solubilizing or fluidizing the cells’ wall lipid components, allowing for a higher cellular uptake [93,95]. Aside from having reported higher drug permeation and deposition, these vesicles have also been reported to have a higher drug entrapment efficiency than their ancestor’s liposomes, higher stability (prevention of drug degradation via temperature, oxidation, or even light), and a relevant controlled release capacity [84,92,94]. Common preparations methods include the thin-film hydration method, the vortexing method, the modified handshaking method, the suspension homogenization method, the centrifugation method, the reverse-phase evaporation method, the high-pressure homogenization method, and the ethanol injection method, all of which are commonly followed by sonication in order to reduce particle size and increase its homogeneity [93,94,95]. They have become an important tool in the successful encapsulation of not only small molecular weight hydrophobic and hydrophilic drugs, but also larger molecules, such as nucleic acids, proteins, or peptides [84,92,95].

Finally, niosomes, the most recently developed technology to overcome the disadvantages of liposomes as drug delivery systems, are likewise nanometric lamellar structures, mainly composed of a lipid (mostly cholesterol) and a non-ionic surfactant [96,97,98]. Nevertheless, unlike what happens in liposomes, in niosomes, cholesterol has a support function, shaping the vesicles and providing stiffness to their structure, while the main constituent of the bilayer is the surfactant, namely amphiphilic non-ionic surfactants, such as sorbitan esters (Span), polysorbates (Tween), or others [84,96,97]. Niosomes can be generally classified as either small unilamellar (one bilayer, between 10 nm and 100 nm), large unilamellar (one bilayer, larger than 100 nm), or multilamellar (more than one bilayer, larger than 50 nm) [96,98,99]. While smaller unilamellar vesicles have the advantages of being typically more stable (high thermodynamic instability) and easier to obtain, larger vesicles, and/or with a higher number of layers, have the ability to encapsulate a higher amount and/or wider variety of drugs (more than two drugs simultaneously, and/higher amounts of each one) [96,99]. Common preparation methods include the thin-film hydration method, the ether injection method, the reverse-phase evaporation method, the microfluidization method, the transmembrane pH gradient method, the bubble method, the supercritical carbon dioxide fluid method, the heating method, or the ball milling method, all typically followed by sonication whenever necessary to reduce particle size and increase its homogeneity [84,96,97]. Niosomes have the advantages of reportedly being more stable, having a higher entrapment efficiency, and having a more controlled drug release capacity and targeted delivery capability than liposomes [96,98]. Other reported advantages of these nanosystems are their biocompatibility, biodegradability, low cost, sustainability, and easy scale-up [97,98,99]. The increased permeation that results from drug encapsulation inside this type of vesicles is mostly due to the permeation-enhancing capability of the used surfactants [84,96,99]. These vesicles have been developed for the treatment of several types of diseases, such as cancer (breast, lung, prostate, and cervical), cardiovascular complications (ischemic heart disease and myocardial ischemia), infections (HIV and hepatitis B), rheumatoid arthritis, diabetes, and neurological disorders [96,98,99].

Nevertheless, although several literature reviews already exist on liposome-derived nanosystems for administration on (topical) or through (transdermal) the skin [100,101,102], or for diseases such as cancer or fungal infections [103,104,105], so far, none have focused on the relevance of this type of vesicle for the treatment of diseases with a brain etiology. Hence, the purpose of this review was to highlight the significant potential of ethosomes, transfersomes, and niosomes for the treatment of high impact and high prevalence psychiatric and neurodegenerative diseases, such as schizophrenia, bipolar disorder, depression, anxiety, Alzheimer’s disease, and Parkinson’s disease. This sought out critical approach aimed to highlight the relevance of these vesicles for a more effective and safer treatment of these diseases, including the analysis of relevant data, such as particle size, polydispersity index (PDI), zeta potential (ZP), encapsulation efficiency (EE), in vitro drug release, ex vivo drug permeation, in vitro efficacy and safety, and in vivo pharmacokinetic and pharmacodynamic performance. A summary of the most relevant information regarding each study is given in Table 1.

## 3. Liposome-Derived Nanosystems: Ethosomes, Transfersomes, and Niosomes for Brain Drug Delivery

### 3.1. Ethosomes

Ligustrazine is a natural-derived compound used in traditional Chinese medicine for the treatment of several different cardiovascular and cerebrovascular diseases. Recently, some studies have shown that this alkaloid, extracted from *Ligusticum chuanxiong* Hort (*Haoben Chuanxiong*), effectively has neuroprotective effects, related to its marked antioxidant, anti-inflammatory, anti-apoptotic, calcium overload inhibition, and enhanced hippocampal cholinergic system function effects, resulting in learning and cognitive function improvements. Nevertheless, when administered orally, this compound leads to variable drug absorption, low bioavailability, and a short elimination half-life due to extensive hepatic first-pass metabolism, which makes frequent administration necessary. Furthermore, its intravenous administration, aside from having the general disadvantages associated with invasive administrations, requires infusion for a long period of time, leading to reduced patient compliance and both local and systemic side effects [106,117,118,119]. To tackle these issues, Shi et al. [106] developed ethosomes, made of ethanol and egg phosphatidylcholine, to encapsulate ligustrazine for transdermal delivery for the treatment of Alzheimer’s disease. Just like intravenous administration, transdermal drug delivery aims to deliver drugs to the systemic blood circulation, but instead of the drugs being injected directly into the blood vessels, they reach them after permeating the skin, where the formulation is applied [120,121]. This leads to the obvious advantage of non-invasiveness, hence increasing patient compliance, while also being an alternative to the oral route, in cases when this route is not available, and also allowing the avoidance of gastrointestinal, hepatic chemical, and metabolic drug degradation and being able to provide sustained plasma drug concentrations [122,123]. Despite the associated advantages, transdermal delivery faces the challenge of actually being able to make the drug permeate the skin barrier, which can be quite difficult [124,125]. Here, nanosystems, and more specifically ethosomes, can be of help, due to their small size and permeation-enhancing capability. In the case of this study, by Shi et al. [106], the ethosomes were produced using the classical cold method, followed by sonication for particle size reduction, and then incorporated into a carbomer gel matrix. Small particle sizes (146.3 ± 24.6 nm), reduced PDI values (0.034 ± 0.009), a reasonably high EE (70.23 ± 1.20%), and a pH of 5.9 revealed the adequacy of the developed formulation for skin application. Furthermore, these ethosomes were revealed to be stable for at least 4 weeks after preparation, while stored under refrigeration, by only showing a small increase in particle size (2.2 ± 0.4%). Moreover, the in vitro skin permeation assay (rat skin, performed on Franz cells) showed an increased skin depth penetration of the ethosomal gel, when compared to a drug solution or ethanolic carbomer gel, having an overall higher drug permeation. Additionally, in vivo pharmacodynamic experiments, in rats, using the Morris water maze test, with chronic transdermal administration of the developed preparations (for 9 days), showed that the ethosomal gel reversed scopolamine-induced memory deficits completely, with a decrease in escape latency down to the levels of healthy animals. In addition, biochemical estimations of the relevant antioxidant enzymes proved that the developed formulation led to a complete recovery of brain superoxide dismutase and glutathione peroxidase activity (reduced via prior scopolamine administration), thus revealing that the developed ethosomes might not only lead to an improvement in memory deficits provoked by Alzheimer’s disease but could also contribute to slowing down the disease’s progression by reducing oxidative stress. Given these promising results, the developed transdermal ethosomal gel could have great potential as an alternative for the treatment of Alzheimer’s disease.

### 3.2. Transfersomes

Curcumin is another natural-derived and thoroughly studied compound, extracted from the rhizomes of *Curcuma longa* L. It has been described to have substantial anti-inflammatory and antioxidant effects, which are mostly connected to the polyphenols that are part of its composition. It has also been proven to have neuroprotective properties in Alzheimer’s disease, namely by targeting neurotrophins and cellular processes connected to cytokine production [107,126,127,128]. On the other hand, berberine, a natural-occurring alkaloid extracted from *Berberis* species, which has long been used in traditional Chinese medicine, has also been demonstrated to have neuroprotective effects, namely through inhibition of the apoptosis-inducing Akt/ERK1/2 signaling pathway, c-Jun N-terminal kinase pathway, and GSK-3β and caspase-3 activity [107,129,130,131]. Nevertheless, despite the strong suggestion of beneficial effects in the treatment of neurodegenerative diseases, both compounds have limited efficacy due to low solubility and low bioavailability. To address these issues, Mishra et al. [107] decided to simultaneously formulate both molecules in transfersomes, for intranasal administration, for the treatment of Alzheimer’s disease. Aside from being non-invasive, easily appliable, and a good alternative to the oral route when it is not available, intranasal administration has the major advantage of allowing for at least part of the drug to be transported directly from the nasal cavity to the brain, via neuronal pathways, a process known as nose-to-brain drug delivery [132,133]. This makes it ideal for brain diseases, since the drug will be transported to the intended therapeutic site of action without undergoing first-pass hepatic metabolism, and without having to pass through the BBB, which is known for having a very low level of permeability to most molecules [134,135]. Hence, intranasal delivery will not only allow an increase in brain drug bioavailability, but also a decrease in systemic drug distribution, making it both a more effective and safer therapeutic alternative, while also having a faster onset of action than other administration routes, making it ideal for managing emergency situations [136,137]. In the study by Mishra et al. [107], three different transfersomes were produced, containing the drugs separately or together. All vesicles were produced using the film hydration method, followed by sonication, and composed of L-α phosphatidylcholine and sodium deoxycholate. Particle size varied from approximately 130 nm to 170 nm, with a PDI from 0.054 to 0.120, ZP from around −7 mV to −32 mV, and EE from 65% to 68%, showing that the vesicles were not only small and homogeneous in size, but also had the potential of good stabilization due to electrostatic repulsion, with a reasonably high encapsulation capacity. Additionally, in vitro drug release studies (dialysis bag method) in acidic and alkaline media (representative of the hippocampus and cerebrospinal fluids, respectively) showed that the vesicles released both drugs in a sustained manner, with berberine achieving an overall higher cumulative drug release than curcumin. Moreover, the release of both drugs was higher under acidic conditions (when compared to alkaline ones), and the co-formulation of both drugs increased the release of curcumin (compared with vesicles only containing curcumin). This might contribute to a more prolonged therapeutic effect, especially in the hippocampus (acidic pH), with consequently minimized side effects. Furthermore, all transfersomes showed high acetylcholinesterase inhibition activity, especially the dual-loaded vesicles, being comparatively better than the free drugs, possibly due to enhanced cellular penetration of the developed vesicles, as well as synergy between curcumin and berberine. Other relevant biochemical parameters were determined, with decreased malondialdehyde and nitric oxide concentrations and reduced superoxide dismutase and catalase activities being most significant for the developed vesicles (when compared to controls), and especially the curcumin–berberine transfersomes, showing once more the potential of these nanosystems in this case in what concerns oxidative mechanisms related to the pathogenesis of Alzheimer’s disease. Additionally, the developed transfersomes proved to be able to effectively reach the brain after intranasal administration in mice, since they led to a higher brain maximum drug concentration (C_max_), overall drug exposure (represented by the area under the curve (AUC) values), and retention (represented by the mean residence time (MRT) values), with the dual-loaded vesicles once again leading to the most promising results, hence indicating synergistic effects once more. Furthermore, the developed formulations proved to have therapeutic-like effects in an Alzheimer’s disease animal model (scopolamine-induced), since chronic administration through the intranasal route led to improved spatial memory and locomotor activity, with the best results being again reached with the curcumin–berberine vesicles. Additionally, the developed transfersomes were also demonstrated to be safe for administration, since the results of a hemolysis study, which were obtained using the classical assay for hemocompatibility and acute toxicity assessment, showed that these vesicles led to much lower hemolysis percentage values than the free drugs, indicating that a higher safety resides in these molecules’ encapsulation. Hence, brain-targeted transfersomes co-loaded with curcumin and berberine were successfully prepared, proving to be a potentially promising platform for new therapeutic options for the treatment of neurodegenerative diseases, such as Alzheimer’s disease.

Recent studies have also proven that insulin can have a potentially beneficial role in the treatment of Alzheimer’s disease, leading to improved cognitive function, with neuroprotective effects being mainly due to action against oxidative stress, inflammation, and mitochondrial damage related to the PI3K/Akt and MAPK signaling pathways. Additionally, insulin has proven to increase glucose metabolism in brain cells, lead to the changing of Aβ oligomers’ ratio, and increasing brain high-energy phosphate content, all of which have an important role in the disease’s pathophysiology. Nevertheless, being a protein, it is hard to deliver insulin to the brain, especially due to its high molecular weight, leading to low permeability through the BBB, and also high susceptibility to degradation [108,138,139,140]. Hence, in order to deliver insulin with high efficacy to the brain, for the treatment of Alzheimer’s disease, Nojoki et al. [108] encapsulated it inside transfersomes for intranasal delivery. The nanosystems were produced via thin-film hydration and were composed of soy lecithin (a phospholipid) and Tween 80 (an edge activator). These transfersomes had a particle size, PDI, ZP, and EE of 95.2 ± 19.0 nm, 0.265, −3.5 mV, and 69.6 ± 1.2%, respectively. Then, a variation of these transfersomes was made by modifying their surface with chitosan. The chitosan-coated transfersomes showed a slightly bigger particle size (137.9 ± 28.2 nm), but smaller PDI (PDI 0.20), and a similar EE (65.1 ± 0.9%). The coated transfersomes also presented a higher and positive ZP (+23.4 mV), suggesting that the coating with chitosan, a positively charged polymer, was indeed successful. The developed particles had a spherical shape and proved to be reasonably stable for up to 3 months, under storage at 4 °C and 25 °C, since the particle sizes did not change significantly, hence suggesting that no aggregation phenomenon occurred, although there was a reduction in encapsulation efficiency. In vivo experiments, conducted in rats, showed that a higher drug targeting and retention in the brain was achieved with the chitosan-coated transfersomes, as opposed to the rest of the body, and when compared to a drug suspension, indicating a higher bioavailability at the intended therapeutic site, and thus reducing systemic drug exposure (Figure 3A). The positive charge of the coated particles, which is due to chitosan’s protonated amine groups, could have been a beneficial parameter in mucoadhesion, since they could have formed electrostatic bonds with the negatively charged nasal mucosa, hence enhancing drug absorption and, consequently, improving drug bioavailability. Additionally, chitosan is also known to lead to a temporary opening of tight junctions, which could have led to a further improvement in nasal drug absorption. The beneficial role of the chitosan coating in brain drug targeting was further supported by the pharmacodynamic study results in an Alzheimer’s disease rat model (which involved the Morris water maze test and pre-treatment with streptozotocin). The study showed that the intranasally administered chitosan-coated transfersomes led to a more significant reduction in escape latency time, and an increase in the swimming speed, distance traveled, and time spent in the target zone than all other groups (intravenous or intranasal saline, intranasal drug suspension, or intranasal-uncoated transfersomes), leading to the most substantial improvements in movement, learning, and memory performance (Figure 3B–E). Additionally, a histological evaluation in the rat’s brain tissues showed that insulin administration led to an increase in the number of pyramidal cells, a decrease in neuronal loss and atrophy in the hippocampus, and a general increase in the number of healthy neurons, with these positive morphological changes being most noticed after the intranasal administration of the chitosan-coated transfersomes. Hence, the developed insulin chitosan-coated transfersomes proved to lead to increased brain drug delivery through the intranasal route, with decreased systemic distribution and good neuroregeneration capability, making them a promising alternative for future Alzheimer’s disease treatments.

Another study, by ElShagea et al. [109], also evaluated the potential of transfersomes for nose-to-brain drug delivery. The encapsulated drug was rasagiline, a monoamine oxidase inhibitor used for the treatment of Parkinson’s disease, which has low oral bioavailability due to having extensive hepatic first-pass metabolism [109,141,142,143]. The vesicles were produced using the thin-film hydration technique, followed by sonication, and were composed of phosphatidylcholine (a phospholipid) and sodium deoxycholate (an edge activator). After production, the transfersomes were incorporated into an in situ gel made of pectin, a mucoadhesive natural polysaccharide with gelling capacity, and a mixture of poloxamers, namely Pluronic^®^ F-127 and Pluronic^®^ F-68, which are thermosensitive polymers with an additional gelling ability. The developed spherically shaped vesicles (Figure 4A,B) revealed a particle size of 198.635 ± 34.98 nm, a PDI of 0.45 ± 0.079, a ZP of −33.45 ± 4.73 mV, and an EE of 95.735 ± 0.091%. Additionally, in vitro drug release (dialysis cellulose bag method, Figure 4C) revealed a dual-peaked drug release profile for the transfersomes in aqueous suspension, with half of the drug undergoing a burst release within the first half-hour, followed by a sustained and complete release for up to 8 h. Nevertheless, the incorporation of the vesicles into the in situ gel eliminated the initial burst release, revealing a more balanced and controlled drug release profile, probably due to an enhanced formulation viscosity. Furthermore, the developed preparation was also revealed to be stable for up to 3 months, showing no visible sedimentation or particle aggregation after storage under refrigeration, and also having led to no significant changes in the particle characterization parameters (size, PDI, ZP, and EE). In vivo pharmacokinetics, in rats, comparing the developed intranasal transfersomal in situ gel with an intravenous drug solution (Figure 4D), showed that the brain C_max_ and AUC values were significantly higher for the in situ gel, reaching a much higher brain bioavailability with no delay (C_max_ achieved at the same time (T_max_) as intravenous administration). Additionally, calculated drug targeting indexes showed that the intranasal formulation not only succeeded in increasing brain drug bioavailability, as opposed to its plasma concentration, but probably did it in a direct manner, making use of the neuronal nose-to-brain transport pathways. Aside from the positive contribution of the small size of the developed vesicles, these results could also be partially due to the mucoadhesive properties of pectin and the gelling and viscosity-increasing properties of both pectin and poloxamers, which allowed the formulation to have a longer retention time in the nasal mucosa, hence increasing the time available for drug absorption to occur. Additionally, it was suggested by the authors that Pluronic^®^ F-127 might have interacted with the nasal mucosa itself, via entanglement glycoprotein chains, leading to a further reduction in mucociliary clearance and, consequently, a higher drug absorption. The biocompatibility and, hence, potential safety of the developed formulation was also evaluated, resorting to a histopathological study in rat nasal mucosa (Figure 4E,F). The results indicated an overall absence of nasal tissue damage, with no signs of hemorrhage or necrosis, and with the epithelial layer remaining intact, with unaltered basal membrane and submucosa, deeming the preparation safe for intranasal administration. Thus, a novel intranasal transfersomal formulation was successfully developed, showing promise for direct nose-to-brain drug targeting for the treatment of Parkinson’s disease.

Aripiprazole is another drug molecule approved for the treatment of diseases with a brain etiology, namely as a main therapy for schizophrenia and bipolar disorders, and as an adjuvant therapy for major depressive disorders. It is characteristically categorized as an atypical antipsychotic drug, acting as dopamine D2 and D3 and serotonin 5-HT1A receptor partial agonists, and as a serotonin 5-HT2A receptor antagonist. Aripiprazole is mainly administered via the oral and parenteral routes, but due to lack of brain selectivity, these formulations lead to several systemic side effects, some of them being quite severe, such as hypotension, somnolence, akathisia, tremors, or neuroleptic malignant syndrome. Aside from these safety issues, which directly affect patient compliance, being a hydrophobic drug not only leads to the difficulty of formulating aripiprazole with high strength but also results in a high variability of blood levels and, consequently, a variable and unpredictable therapeutic response. This drug is also a P-gp substrate, which limits its entry into the brain due to BBB efflux, limiting the amount of drug that is capable of reaching the intended therapeutic site of action [144,145,146,147]. Therefore, in order to solve these problems, and have a more effective and safer aripiprazole administration, Taymouri et al. [144] encapsulated this drug into transfersomes for intranasal administration. The vesicles, produced via thin-film hydration followed by sonication, were made of soybean lecithin (a phospholipid) and sodium deoxycholate (an edge activator), and then incorporated into a gellan gum hydrogel base for delaying mucociliary clearance and increasing the formulation’s contact time with the nasal mucosa. Gellan gum is a biocompatible exocellular deacetylated bacterial polysaccharide, with an anionic nature, and hence can form an in situ gel when in contact with the numerous cations present in the nasal mucosa (mainly potassium, calcium, and sodium), since the complexation with those cations leads to the formation of a three-dimensional network. The developed transfersomal in situ gel was characterized, revealing a small particle size (72.12 ± 0.72 nm) and PDI (0.19 ± 0.07), a high negative ZP (−55.6 ± 1.9 mV), and a good EE (97.06 ± 0.10%). In vitro drug release assays (dialysis bag method) showed that both the transfersomes in aqueous suspension and the transfersomal in situ gel had a biphasic drug release, with an initial burst release followed by a sustained release profile. Nevertheless, the incorporation of the transfersomes into the in situ gel decreased the burst release and led to a more controlled release profile, since the drug has to partition from the transfersomes into the gel first, and only after that can it diffuse through the gel matrix into the dissolution media, with the high viscosity of the gel matrix also having an important role in this matter. The potential of the developed formulation was confirmed through in vivo pharmacodynamic tests, in mice with ketamine-induced psychosis, which showed that the reductions in locomotor activity and immobility, swimming, and climbing times were most significant after intranasal administration of the developed aripiprazole transfersomal in situ gel when compared to controls (no treatment or intranasal, oral, or intraperitoneal drug solutions). The obtained promising results of the developed formulation in brain-targeted delivery for the treatment of schizophrenia and derived disorders could be attributed to the lipophilic nature, nanometric size, and high membrane flexibility of the transfersomes, which could have been an important contributing factor for direct brain drug delivery through the neuronal pathways, but also to the excipients that were part of vesicle composition, soybean lecithin and sodium deoxycholate, which both act as permeation enhancers. Additionally, the incorporation of the transfersomes into the gellan gum matrix could have produced better results by providing a prolonged therapeutic effect, due to high viscosity and mucoadhesion.

On the other hand, asenapine, an atypical antipsychotic drug which mainly acts on dopamine D2 and serotonin 5-HT2A receptors as an antagonist, has low oral bioavailability, mainly due to liver and gut metabolism [110,148,149,150]. Hence, to solve this issue, Shreya et al. [110] encapsulated asenapine in transfersomes, for transdermal delivery, for the treatment of schizophrenia and bipolar disorder. The vesicles were once more prepared via thin-film hydration, followed by sonication, and were also made of soy phosphatidylcholine and sodium deoxycholate. Nevertheless, in this study, the transfersomes were then incorporated into an ethanolic Carbopol 934P gel, leading to a particle size of 126.0 nm, PDI of 0.232, ZP of −43.7 mV, and EE of 54.96%. Ex vivo permeation results (rat skin, Franz diffusion cells) showed that the transfersomal gel led to increased drug permeation, with an evident synergy existing between the used nanotechnological (transfersomes) and chemical (ethanol) permeation enhancement approaches. In the skin, ethanol will dissolve some of the stratum corneum’s lipids, transiently disrupting the skin barrier and leading to enhanced drug permeation. An in vivo pharmacokinetic study in rats, comparing the developed transdermal transfersomal gel to an oral drug solution, further supported the relevant potential of the developed formulation, since the transfersomal gel led to higher C_max_ and AUC values, but also T_max_ values, meaning that the transfersomes resulted in an increased and prolonged bioavailability. The elimination parameters also additionally supported these conclusions, since the transdermal transfersomal gel led to a higher elimination half-life (t_1/2_) and MRT than the oral solution, hence depicting a longer retention of the drug in the body. Thus, the developed transdermal asenapine transfersomal gel presented an overall relevant potential as an alternative to oral conventional formulations, for the treatment of schizophrenia, proving that the space for improvement in the treatment of these diseases could in fact be filled with this type of approach.

### 3.3. Niosomes

As mentioned, asenapine is a dopamine and serotonin antagonist with low bioavailability. Hence, just like Shreya et al. [110], Singh et al. [111] developed vesicles to encapsulate this drug, for increased brain targeting, for the treatment of schizophrenia and bipolar disorder. Nevertheless, in this study, instead of transfersomes, niosomes were produced for oral administration. Oral drug administration is still the go-to administration route for most situations, due to being non-invasive, hence not bringing the patient pain or even discomfort, making it best for chronic therapies, and being easy to self-administer, leading to high patient compliance [151,152]. Additionally, it is possible to reach a prolonged therapeutic effect due to modification of the pharmaceutical oral forms due to controlled drug release, and the drug has access to a large area available for absorption to occur [153,154]. Furthermore, although the harsh environment of the gastrointestinal tract can lead to chemical and metabolic drug degradation, and even if the BBB has very low permeability to most drugs, nanoformulations can be designed to not only protect drugs, but also increase their permeation to the brain tissues [155,156]. In this context, the niosomes developed by Singh et al. [111] were made of cholesterol and Span 60 using the organic solvent injection method. The vesicles were spherically shaped and had a quite small (84 ± 5 nm) and homogeneous (PDI 0.27) particle size, with a ZP of −17.53 mV, and an EE of 70%. In vitro drug release profiles (dialysis bag method) showed an initial burst release, probably of the part of drug that was adsorbed on the surface of the particles, followed by a sustained release, of the part of the drug that was effectively encapsulated. After 8 h, the cumulative drug release reached 68%, being considered a reasonably high amount, with the release profile showing controlled release characteristics. An in vivo pharmacodynamic study in rats (ketamine-induced psychosis, open field chamber), where the developed niosomes were compared to a drug solution, both administered orally, showed that the niosomes led to a significantly improved locomotor activity, with the behavioral response being similar to the control group (no induced disease) and performing better than that of the drug solution. Furthermore, an in vivo pharmacokinetic study further supported the potential of the developed vesicles, since the asenapine-loaded niosomes led to higher C_max_, AUC, and t_1/2_ values than the drug solution. Hence, the developed niosomes led to an improved bioavailability and therapeutic response, being a promising alternative for the oral administration of asenapine, for the treatment of schizophrenia, bipolar disorder, and related psychotic disorders.

Olanzapine is another atypical antipsychotic drug molecule, acting mainly on dopamine D2 and serotonin 5-HT2A receptors, which has low oral bioavailability due to extensive first-pass metabolism, also having low water solubility, making it a good candidate for encapsulation into niosomes [112,157,158,159]. This was exactly what Khallaf et al. [112] conducted, incorporating olanzapine into cholesterol and Span 80 niosomes for intranasal administration and for the treatment of schizophrenia and related psychotic disorders. The vesicles were produced via the thin-film hydration technique, followed by sonication. A variation of these niosomes was also made, by coating them with chitosan, due to this polymer’s bioadhesive properties, making it prone to interact with the nasal mucosa (more specifically with mucin), and also due to its additional ability to enhance drug permeation (transient opening of tight junctions). Both the uncoated and coated niosomes had a spherical shape and had a particle size of 241.30 nmand 250.1 ± 5.0 nm, an EE of 71.2% and 71.9%, and a viscosity of 3.1 ± 0.9 cP and 8.4 ± 1.2 cP, respectively. Hence, chitosan coating of the niosomes did not lead to significant changes in drug encapsulation, slightly increased the formulation’s viscosity, and also led to a small increase in particle size, which confirmed that the coating was in fact successful. The developed vesicles were also stable (6 months, 4 °C), showing only a small and statistically insignificant increase in particle size and decrease in EE. Ex vivo permeation assays (sheep nasal mucosa, Franz diffusion cells) revealed that the developed niosomes led to a better drug permeation through the nasal mucosa than a drug solution, probably in part due to the presence of a non-ionic surfactant in the vesicles’ membrane, an excipient type that is a known permeation enhancer. Additionally, the chitosan-coated niosomes led to a higher drug permeation than the uncoated ones, likely due to chitosan also being a permeation enhancer through tight junction transient openings, and also due to its mucoadhesive properties, leading to longer formulation retention at the absorption site. Moreover, confocal laser microscopy images of the nasal mucosa treated with fluorescence-labeled niosomes confirmed that the depth of penetration was higher for the coated niosomes, when compared to the uncoated ones, being statistically significant. The coated vesicles’ mucoadhesion was confirmed experimentally since they revealed a substantial mucoadhesive strength (42 ± 3.2 dyne/cm^2^). An in vivo pharmacokinetic study, in rats, showed that the brain C_max_ and AUC values were higher for the intranasally administered coated niosomes than that for an intranasal drug solution. Additionally, the intranasally coated niosomes led to a longer drug retention in the animals’ body, leading to a higher t_1/2_ and MRT than the intranasal and intravenous drug solution groups. Hence, the intranasally administered surface-modified olanzapine-loaded niosomes had a good brain targeting effect, resulting in substantial brain drug delivery, and leading to a sustained effect due to longer drug retention. The safety of the developed formulation was also confirmed via histopathological studies, with the rats’ olfactory region histological structure showing no signs of irritation or tissue damage (such as cellular deformation, edema, hemorrhage, or necrosis). Hence, olanzapine-loaded chitosan-coated niosomes, delivered via the intranasal route, could become a promising alternative to the oral route for the treatment of schizophrenia and related psychiatric disorders, offering effective brain targeting, escape from first-pass metabolism, and possible required therapeutic dose reduction.

Rivastigmine is an acetylcholinesterase and butyrylcholinesterase inhibitor used for the treatment of Alzheimer’s disease, leading to a reduction in the cognitive decline associated with cholinergic neuron degeneration. Nevertheless, it has extensive first-pass metabolism, leading to low oral bioavailability, and a short-half life, leading to the need for frequent administration [113,160,161,162]. On the other hand, N-Acetyl cysteine has also proven to have beneficial properties in neurodegenerative diseases, since it has been shown to increase glutathione levels, leading to an increased depletion of reactive oxygen species and, consequently, aiding in preventing the inflammation that is related with neuronal damage [113,163,164,165]. Therefore, in order to tackle the issues related with rivastigmine bioavailability and take advantage of the neuroprotective properties of N-Acetyl cysteine, Kulkarni et al. [113] decided to develop intranasal niosomes encapsulating both compounds for the treatment of Alzheimer’s disease. The vesicles were prepared using the ethanol injection method, followed by high-speed homogenization, and were composed of cholesterol (a lipid) and Span 20 (a non-ionic surfactant). The niosomes had a spherical shape and small nanometric size (162.7 nm) (Figure 5A,B), with high homogeneity (PDI < 0.1), and a negative ZP (−24.8 mV). EE values were high for both active compounds, being 97.7% for rivastigmine and 85.9% for N-Acetyl cysteine. Additionally, the formulation proved to be stable under refrigeration (from 4 °C to 8 °C) since there were no significant changes in either particle size or EE for 6 months. In vitro drug release assays (reverse dialysis method) showed that both compounds were released from the niosomes in a sustained manner, reaching >95% after 48 h, under a diffusion-controlled mechanism, fitting a first-order kinetic model. Ex vivo permeation studies (sheep nasal mucosa, Franz diffusion cells, Figure 5C) showed that the developed dual-loaded vesicles led to a significant drug permeation improvement, both in terms of speed and total amount, when compared to a drug solution, reaching a total of 92% permeation after 48 h. These results could be related to the nanosized vesicles’ extensive area, available for drug diffusion from the formulation and consequent absorption, due to its small size, or due to having a surfactant in their composition, which will lead to permeation enhancement due to tight junctions’ transient opening. In vitro efficacy assays were also performed, in which the developed niosomes achieved a higher acetylcholinesterase and free radical (2,2-diphenyl-1-picrylhydrazyl) formation inhibition than the drug solution, showing relevant anticholinesterase and antioxidant properties, which are both quite relevant in Alzheimer’s disease pathophysiology. These results could be related to an increased contact time of the encapsulated drugs with the cells, and to the fact that drug encapsulation protects them from metabolism, thereby increasing their availability for exerting the required pharmacologic effect. Furthermore, the in vitro hemolysis assay (Figure 5D) proved that the developed vesicles were biocompatible, leading to very low hemolysis values for the drug-loaded niosomes (much lower than for the drug solution), and no hemolysis for the blank niosomes. These promising results were further supported by the in vivo pharmacokinetic study in rats (Figure 5E), in which the intranasal niosomes were compared to an intravenous administration of the same niosomes, and also to an intranasal and intravenous administration of a drug solution. The developed niosomes led to significantly higher AUC, C_max_, t_1/2_, and MRT values than the drug solution, for both routes of administration, with the intranasal delivery of the niosomes overall leading to the most favorable results. Hence, the developed dual-loaded niosomes showed promising results, being a potentially suitable alternative for the treatment of Alzheimer’s disease, with a possible reduction in the dose of rivastigmine due to synergistic combination with N-Acetyl cysteine and showing high versatility due to being possibly suitable for both intravenous and intranasal administration.

Moulahoum et al. [114] also attempted to develop an innovative treatment for Alzheimer’s disease, namely carnosine-loaded niosomes (Figure 6A). Carnosine, which is another name for alanyl-L-histidine, is a natural-derived dipeptide, being present in all mammals, and can be found at its highest concentrations in the brain and muscle tissues, especially in the skeletal and cardiac muscles. Its potential for the treatment of neurodegenerative diseases arises from a substantial level of antioxidant activity (peroxyl radical, oxygen singlet and metal chelation, and related enzymatic regulation), with effective inhibition of advanced glycation end-products, inhibition of amyloid fibril formation, and suppression of β-amyloid accumulation, and protection of brain cells from its cytotoxic effects [114,166,167,168,169]. The niosomes were produced using the thin-film hydration method, followed by sonication, and were made of cholesterol and Span 60. Vesicle morphology was proven to be spherical, with a particle size of 560 ± 203 nm, and an EE of 32.4 ± 5%, with the developed formulation also proving to be stable for up to 30 days under refrigeration. The in vitro drug release assay (dialysis method) showed that the carnosine niosomes had a controlled drug release profile, and in vitro antiglycative and anti-aggregation assays (Figure 6B) proved that the developed vesicles led to decreased amyloid and fibrillation formation in a dose-dependent manner. Furthermore, the niosomes also showed a substantial inhibition of advanced oxidation protein products formation, proving to have considerable antioxidant properties. Additionally, in order to have a deeper understanding of the regions that might be related to the initiation of aggregation processes, the interaction of carnosine with bovine serum albumin was studied through molecular docking (Figure 6C), and the results showed that carnosine’s binding affinity to albumin was substantially higher than a reference antiglycation molecule (aminoguanidine). Hence, carnosine-loaded niosomes were successfully prepared, showing promising properties for the potential treatment of Alzheimer’s disease, namely in what concerns the prevention of its related molecular mechanisms, such as protein oxidation, glycation, and aggregation processes.

Other natural-derived molecules have also been proven to exhibit efficacy in the treatment of neurodegenerative diseases. Ginkgolide B, a diterpene extracted from *Ginkgo biloba*, has been proven to have potential for the treatment of Alzheimer’s disease, since by having an inhibiting role on the platelet-activating factor, it has been shown to protect neuronal cells that were damaged by Aβ accumulation from further harm. Additionally, this compound has also been shown to reduce the apoptosis induced by the Aβ peptide via the brain-derived neurotrophic factor mechanism, and to have reparative and protective effects on Aβ peptide-damaged mitochondria [115,170,171,172]. Moreover, ginkgolide B also appears to be promising in Parkinson’s disease-related pathophysiology factors, since it was proved to regulate the D-glutamic acid pathway, hence affecting glutamic acid and dopamine metabolism, which are connected to the early stages of this disease [115,173,174]. On the other hand, puerarin, an isoflavone extracted from Pueraria genus plants, has also proven to have potentially beneficial effects in both Alzheimer’s and Parkinson’s disease, due to its antioxidant properties (free radical scavenging, and endogenous antioxidant activity increase), regulation of calcium signaling pathways, functional stabilization of amino acid neurotransmitters and maintenance of the dynamic balance between them (excitatory vs. inhibitory), and regulation of neuronal apoptosis and overall cell damage. Nevertheless, despite their promising potential, these molecules have unfavorable physicochemical properties, such as hydrophobicity and high molecular weight, which makes them both challenging to formulate, and difficult to permeate through biological barriers in general, and the BBB in particular, thus limiting their bioavailability. Therefore, to tackle these issues, Zhou et al. [115] developed niosomes encapsulating both ginkgolide B and puerarin simultaneously, for increased and synergistic therapeutic potential for the treatment of Alzheimer’s and Parkinson’s disease, for intravenous administration. Although intravenous drug administration has the clear disadvantage of being an invasive administration route, being linked to causing pain and discomfort to the patient, and needing to be performed by trained professionals (namely in a hospital setting), it is the only administration route that leads to 100% systemic drug bioavailability, since the drug does not have to bypass any type of barrier (chemical, physical, or biological) in order to reach the blood, as it does not need to be absorbed [175,176]. Hence, the formulation is injected directly into the bloodstream, which also makes this the fastest way to reach it and being quite advantageous in the management of emergency or other acute situations [177,178]. The niosomes developed by Zhou et al. [115] were prepared using the thin-film hydration method, followed by high-pressure homogenization, and were made of cholesterol and MYRJ 49 (polyoxyethylene monostearate), and their surface was modified by adding borneol, an aromatic compound with a potential capability of increasing vesicle permeation through the BBB, in order to increase the drugs’ uptake to the brain. The developed vesicles had a particle size of 142.65 nm, a PDI of 0.261, and an EE of 49.90%. An in vivo pharmacokinetic study, in rats, showed that the developed niosomes led to higher brain drug concentrations than a drug solution, both administered intravenously, with higher C_max_, AUC, t_1/2_, and MRT values, leading to a greater bioavailability and longer retention of the drug in the animal’s bodies. Thus, the developed dual-loaded and functionalized niosomes led to improved BBB penetration, showing promising results for the potential treatment of brain neurodegenerative diseases, such as Alzheimer’s and Parkinson’s disease.

Mathure et al. [116] also developed niosomes for brain drug targeting, encapsulating buspirone for the treatment of anxiety disorders. Buspirone is an agonist for serotonin receptors, with low oral bioavailability and a short half-life, due to extensive hepatic first-pass metabolism, requiring frequent dosing, and hence leading to limited patient compliance [116,179,180,181]. Therefore, in order to address these issues, buspirone-loaded niosomes were developed, using the thin-film hydration method followed by ultrasonication, for intranasal delivery. The vesicles were made of cholesterol and Span 60, and later incorporated into a Carbopol 934P plus hydroxypropyl methylcellulose K4M gel, with added benzalkonium chloride to guarantee microbiological adequacy. The niosomes revealed a spherical shape, in the nanosize range (181.9 ± 0.36 nm), having a negative ZP (−15.4 mV), and high EE (87.7 ± 0.66%). Formulation gelling capacity and viscosity were also assessed, with it showing an adequately short gelling time (4 ± 0.230 min) and high viscosity at pH 5–6 (from 2600 ± 0.48 to 7800 ± 0.56 cP), which could allow for a simple and easy instillation in the nasal cavity (fluid preparation), followed by a quick transition into its gel form triggered by the pH increase when in contact with the nasal mucosa. In vitro drug release (USP dissolution apparatus type II, with the dialysis bag attached to the paddle) showed that the formulation had a high cumulative drug release (84.26 ± 0.26%), and ex vivo drug permeation (Franz diffusion cells, sheep nasal mucosal) results showed that it led to a higher permeation after 8 h than a plain Carbopol 934P gel (60% vs. 83%, respectively). Hence, the developed buspirone-loaded niosomal gel revealed its potential for an increased drug absorption through the intranasal administration route, having a superior performance over conventional formulations.

## 4. Liposome-Derived Vesicles: The Future for Brain Drug Delivery?

Given the very low permeability of the BBB to most drug molecules, delivering therapeutics to the brain becomes a significant challenge. Although the grand majority of marketed formulations are conventional formulations, decreased drug bioavailability at the target site and substantial drug distribution to other organs makes these preparations have low efficacy and safety. Nevertheless, as it has been made clear by the analyzed articles, scientists have developed novel alternatives in the nanosize scale, namely ethosomes, transfersomes, and niosomes, that are able to not only protect the drug molecules by encapsulating them, but also take them to the brain in a targeted manner, thereby increasing therapeutic outcomes in animal models. Although the mechanisms through which these vesicles are able to improve BBB penetration remain unclear, their nanosize and lipidic nature are thought to be relevant factors for increased permeation through any kind of biological barrier, since they are able to mimic these membranes’ composition, while being small enough to pass through them [90,182]. Additionally, the active transport of liposome-derived nanosystems to the brain, through transcytosis or receptor-mediated transport, has also been suggested, with binding to molecules such as glutathione or glucose possible playing a major role in vesicle translocation [90,183,184].

This review sheds a light on the true potential of liposome-derived nanovesicles, namely niosomes, transfersomes, and ethosomes, for the treatment of several highly impactful and prevalent neurodegenerative and psychiatric diseases, for which there is still much room for improvement in what concerns treatment efficacy and safety. Although this type of formulation has yet to reach the pharmaceutical market, the analyzed promising results could be the key to increasing the interest of pharmaceutical companies to take these formulations one step further, into clinical trials, in order to assess their true potential in humans. Issues, such as scalability to an industrial scale, should also be examined to make sure that the nanosized vesicles maintain their optimal properties when produced in larger quantities. Hence, although a substantial amount of work is still necessary to make sure that these novel liposome-derived nanosystems are both safe and effective in a real-world application, preliminary laboratory-scale results appear to indicate that these vesicles have much potential in the treatment of diseases with a brain etiology, especially schizophrenia, bipolar disorder, depression, anxiety, Alzheimer’s disease, and Parkinson’s disease.

## 5. Conclusions

Ethosomes, transfersomes, and niosomes have proven to be a suitable alternative to conventional formulations in the treatment of several highly prevalent psychiatric and neurodegenerative disorders, contributing not only to solving formulation issues related to the drug molecule itself, such as low water solubility, but also leading to drug protection against metabolic and chemical degradation, and drug targeting to the intended site of therapeutic action, the brain. This leads to enhanced brain bioavailability, and decreased systemic distribution, with these formulations successfully achieving higher therapeutic efficacy and safety. These nanosized vesicles attain this by increasing drug permeation through biological barriers, on the one hand due to their composition, with the used surfactants being known permeation enhancers, and on the other hand due to the ultra-flexibility of the vesicles, which gives them enough elasticity to be able to pass through small fenestrations in and between cells. Hence, these vesicles have proven to be novel and improved dosage forms, having shown high potential for the treatment of Alzheimer’s disease, Parkinson’s disease, schizophrenia, bipolar disorder, anxiety, and depression, depicting high efficacy in brain neurotransmitter level restoration, brain oxidative status improvement, and improved locomotor activity and/or memory in animal models. As to whether these formulations could be the future for psychoactive drug administration, it will be possible to know with further studies in a clinical trial context.

## Figures and Tables

**Figure 1 pharmaceuticals-16-01424-f001:**
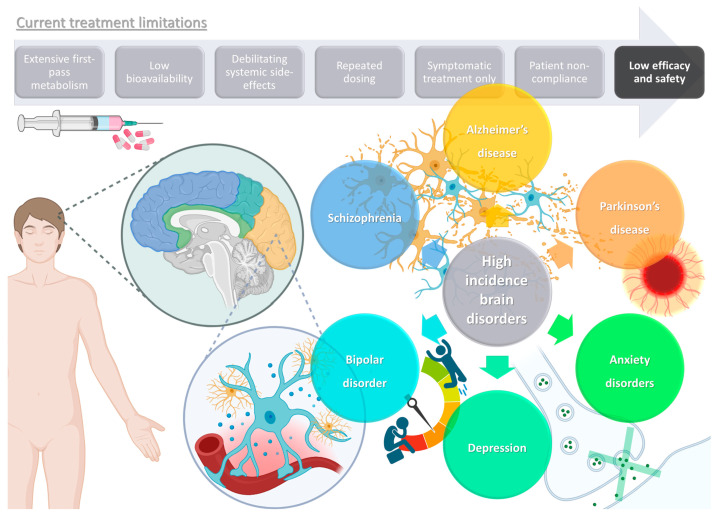
Schematic representation of high-incidence brain disorders and limitations of current treatments (produced with Biorender).

**Figure 2 pharmaceuticals-16-01424-f002:**
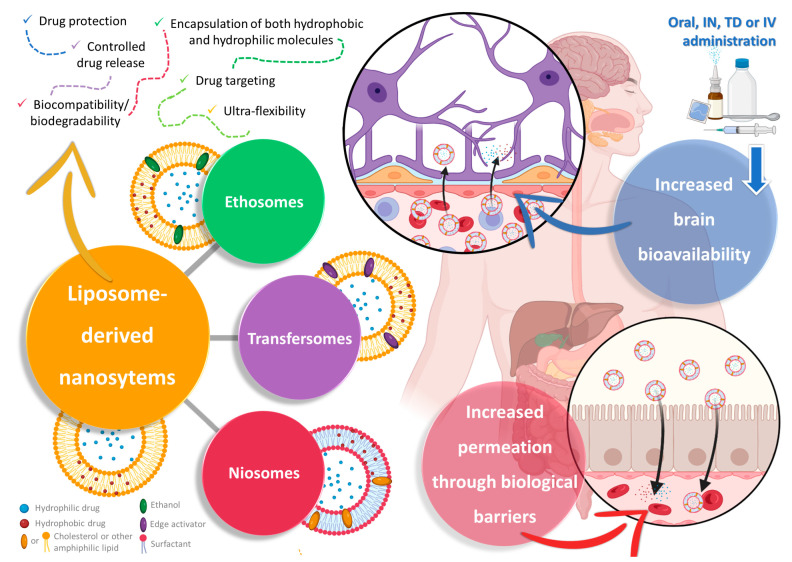
Schematic representation of liposome-derived vesicular nanosystems, namely ethosomes, transfersomes, and niosomes, and their respective composition and advantages in drug delivery (produced with Biorender). IN—intranasal; IV—intravenous; TD—transdermal.

**Figure 3 pharmaceuticals-16-01424-f003:**
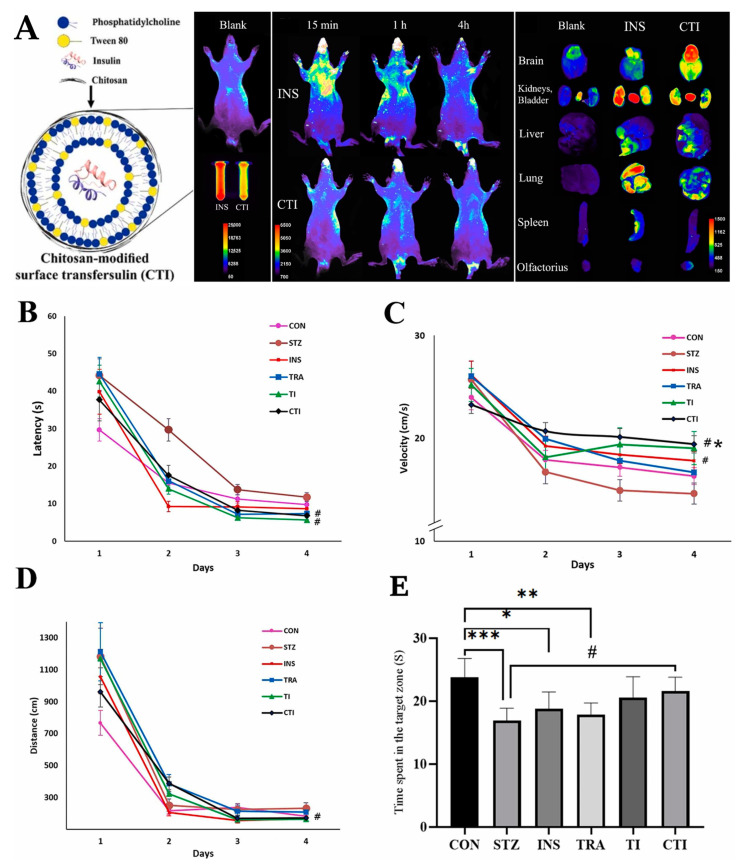
(**A**) Representation of the composition of the developed insulin chitosan-coated transfersomes (left), and optical fluorescence imaging (right) of insulin uptake by several organs after intranasal administration of a drug suspension (FITC-INS) or the developed chitosan-coated transfersomes (FITC-CTI); (**B**–**E**) in vivo pharmacodynamic studies’ results involving the Morris water maze test (and pre-treatment with streptozotocin), in terms of escape latency time (**B**), swimming speed (**C**), distance traveled (**D**), and time spent in the target zone (**E**), after intranasal administration of the developed insulin chitosan-coated transfersomes (CTI), compared to the negative (CON) and positive (STZ) controls, intranasal drug suspension (INS), intranasal-uncoated transfersomes with no drug (TRA), or intranasal-uncoated transfersomes with the drug (TI); # *p* < 0.05 when compared to the control group; * *p* < 0.05, ** *p* < 0.005, *** *p* < 0.0005 when compared to the control group).adapted from Nojoki et al. [108], and reproduced with permission from Elsevier (Creative Commons CC BY 4.0 license).

**Figure 4 pharmaceuticals-16-01424-f004:**
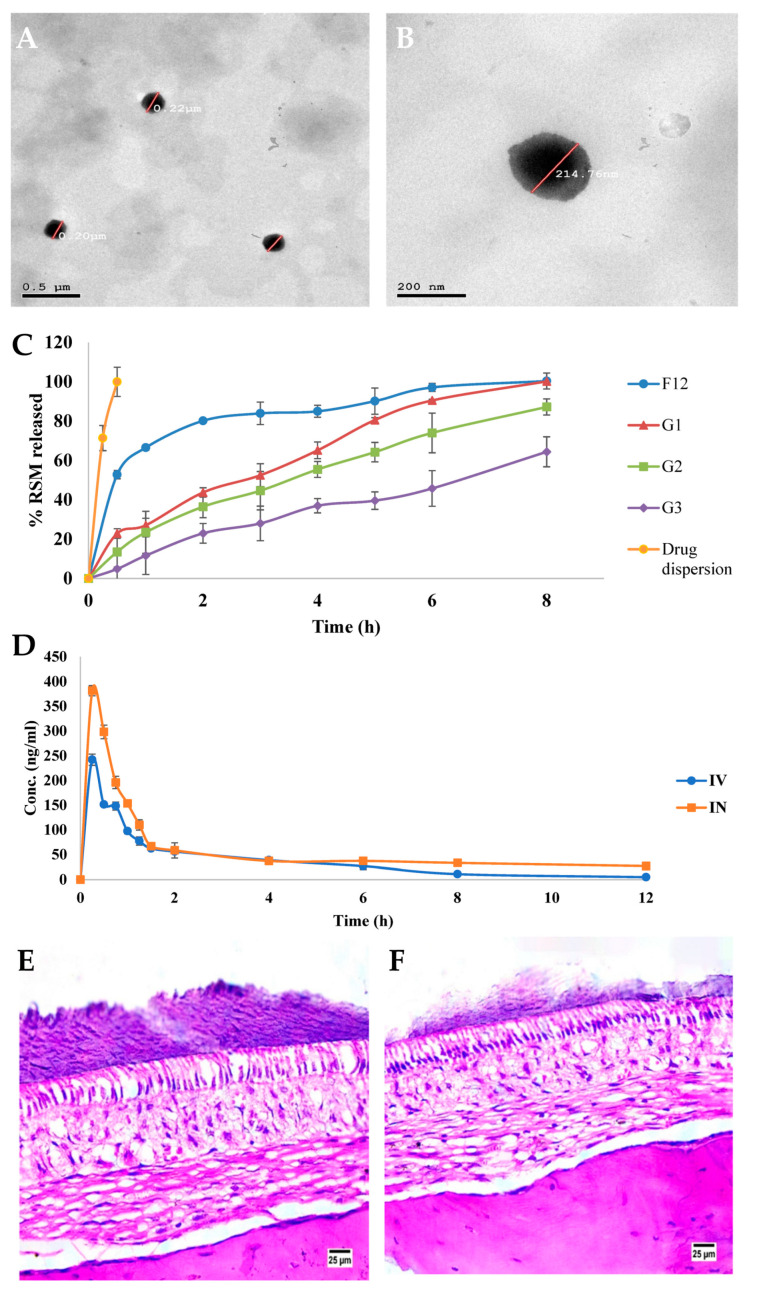
(**A**,**B**) Transmission electron microscopy micrographs of the developed rasagiline transfersomes; (**C**) in vitro drug release profiles of the developed rasagiline transfersomes (F12), transfersomal in situ gels (G1, G2, and G3, with different polymer ratios), and drug dispersion; (**D**) brain drug concentration vs. time curve after intranasal administration of the optimized rasagiline in situ gel, and intravenous administration of a drug solution; (**E**,**F**) histopathological photomicrographs of rat nasal mucosa belonging to an untreated control group (**E**) and after intranasal administration of the optimized rasagiline transfersomal in situ gel (**F**) (hematoxylin and eosin staining); adapted from ElShagea et al. [109], and reproduced with permission from MDPI (Creative Commons CC BY 4.0 license).

**Figure 5 pharmaceuticals-16-01424-f005:**
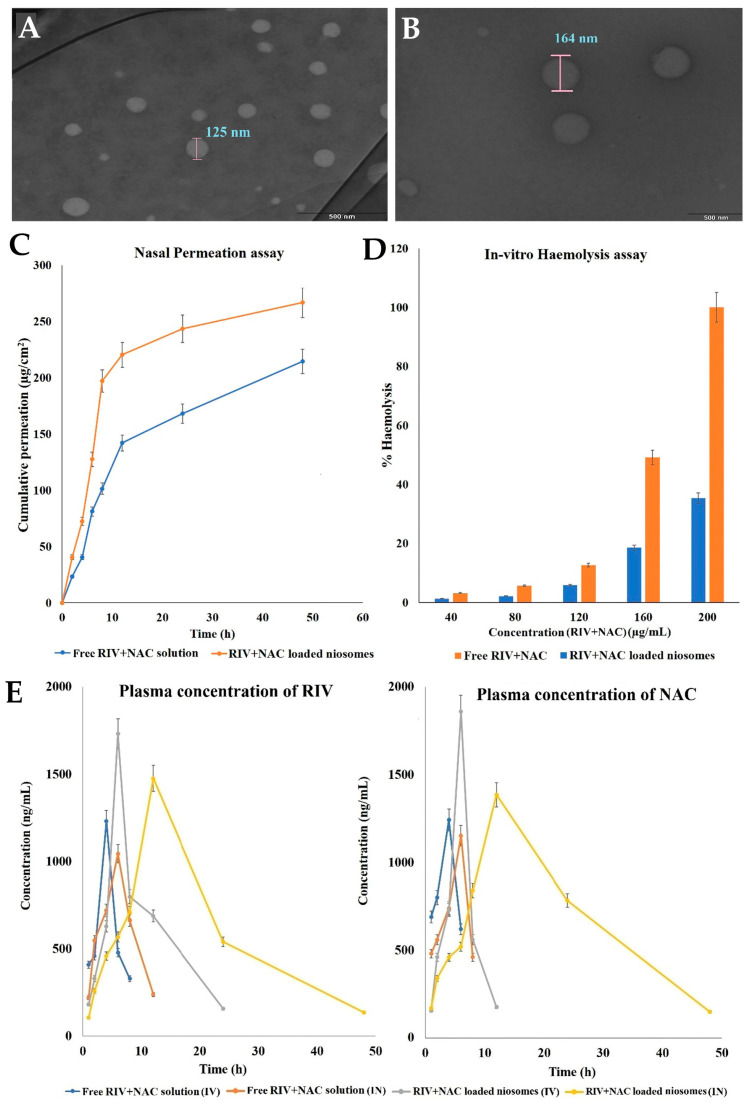
(**A**,**B**) Transmission electron microscopy images of blank niosomes (**A**) and rivastigmine and N-Acetyl cysteine loaded niosomes (**B**); (**C**) cumulative nasal drug permeation of the developed niosomes (RIV + NAC-loaded niosomes) and drug solution (free RIV + NAC solution); (**D**) hemolysis percentage of the developed niosomes (RIV + NAC-loaded niosomes) and drug solution (free RIV + NAC); (**E**) in vivo pharmacokinetics after intranasal administration of the niosomes (RIV + NAC-loaded niosomes (IN)), intravenous administration of the niosomes (RIV + NAC-loaded niosomes (IV)), intranasal administration of a drug solution (free RIV + NAC solution (IN)), and intravenous administration of a drug solution (free RIV + NAC solution (IV)); adapted from Kulkarni et al. [113], and reproduced with permission from Elsevier (license number 5631891480781).

**Figure 6 pharmaceuticals-16-01424-f006:**
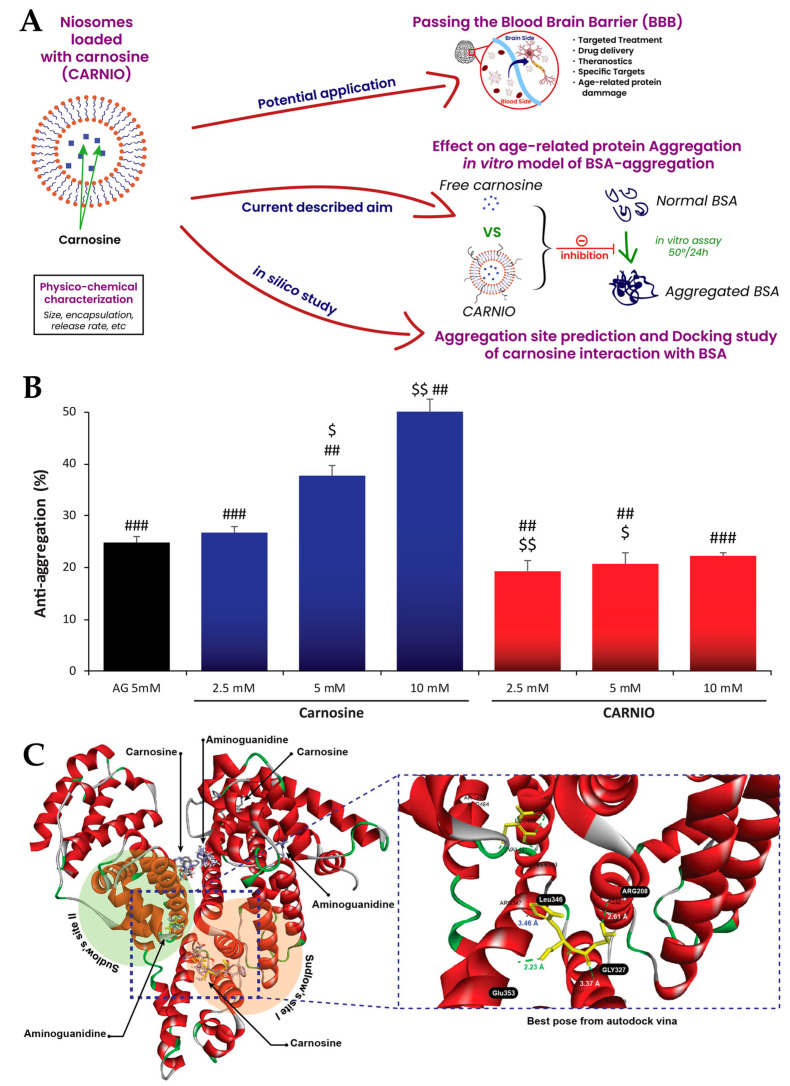
(**A**) Schematic representation of the developed carnosine-loaded niosomes, with potential application in brain targeting for the treatment of neurodegeneration-related molecular mechanisms, and applied assays; (**B**) in vitro anti-aggregation assay results, with estimation of anti-fibril formation effects of free and encapsulated carnosine (CARNIO); (**C**) in silico study of carnosine’s (and aminoguanidine’s) interaction with bovine serum albumin; ## *p* < 0.01, ### *p* < 0.001, when compared to positive control; $ *p* < 0.05, $$ *p* < 0.01, when compared to aminoguanidine 5 mM; adapted from Moulahoum et al. [114], and reproduced with permission from Elsevier (license number 5631900280594).

**Table 1 pharmaceuticals-16-01424-t001:** Summary of the most relevant information regarding each study contained within this review, including the nanosystem type, main formulation composition, encapsulated molecule, disease intended to treat, intended administration route, particle size, polydispersity index, zeta potential, encapsulation efficiency, pH, viscosity, main in vivo pharmacokinetic and/or pharmacodynamic results, and respective reference.

Nanosystem Type	Main Composition	Encapsulated Molecule	Disease Intended to Treat	Intended Administration Route	Particle Size (nm)	PDI	ZP (mV)	EE (%)	pH	Viscosity (cP)	Main PK and/or PD Results	Reference
Ethosomes	Ethanol, egg phosphatidylcholine, and carbomer gel matrix	Ligustrazine	Alzheimer’s disease	Transdermal	146.3 ± 24.6	0.034 ± 0.009	NR	70.23 ± 1.20	5.9	NR	Completely reversed memory deficits and decreased escape latency (rats).	[106]
Transfersomes	L-α phosphatidylcholine and sodium deoxycholate	Curcumin and berberine	Alzheimer’s disease	Intranasal	130 to 170	0.054 to 0.120	−7 to −32	65 to 68	NR	NR	Higher brain C_max_, AUC, and MRT than non-encapsulated drugs, with synergy in dual loading and improved spatial memory and locomotor activity (mice).	[107]
Transfersomes	Soy lecithin, Tween 80, and chitosan	Insulin	Alzheimer’s disease	Intranasal	137.9 ± 28.2	0.20	+23.4	65.1 ± 0.9	NR	NR	Higher brain drug targeting and retention compared to controls, and substantial improvement in movement, learning, and memory performance (rats).	[108]
Transfersomes	Phosphatidylcholine, sodium deoxycholate, pectin, Pluronic^®^ F-127, and Pluronic^®^ F-68	Rasagiline	Parkinson’s disease	Intranasal	198.635 ± 34.98	0.45 ± 0.079	−33.45 ± 4.73	95.735 ± 0.091	NR	NR	Brain C_max_ and AUC values significantly higher than controls (rats).	[109]
Transfersomes	Soybean lecithin, sodium deoxychola, and gellan gum	Aripiprazole	Schizophrenia and bipolar disorders (main therapy); major depressive disorders (adjuvant therapy)	Intranasal	72.12 ± 0.72	0.19 ± 0.07	−55.6 ± 1.9	97.06 ± 0.10	NR	NR	Reduction in locomotor activity and immobility, swimming, and climbing times, higher than controls (mice).	[98]
Transfersomes	Soy phosphatidylcholine, sodium deoxycholate, ethanol, and Carbopol 934P	Asenapine	Schizophrenia and bipolar disorder	Transdermal	126.0	0.232	−43.7	54.96	NR	NR	Higher C_max_, AUC, T_max_, t_1/2_, and MRT compared to controls (rats).	[110]
Niosomes	Cholesterol and Span 60	Asenapine	Schizophrenia and bipolar disorder	Oral	84 ± 5	0.27	−17.53	70	NR	NR	Higher C_max_, AUC, and t_1/2_ values than controls, and significantly improved locomotor activity (rats).	[111]
Niosomes	Cholesterol, Span 80 and chitosan	Olanzapine	Schizophrenia and related psychotic disorders	Intranasal	250.1 ± 5.0	NR	NR	71.9	NR	8.4 ± 1.2	Higher brain C_max_ and AUC, t_1/2_, and MRT (rats).	[112]
Niosomes	Cholesterol and Span 20	Rivastigmine and N-acetyl cysteine	Alzheimer’s disease	Intranasal or intravenous	162.7	<0.1	−24.8	85.9 to 97.7	NR	NR	Higher AUC, C_max_, t_1/2_, and MRT values (rats).	[113]
Niosomes	Cholesterol and Span 60	Carnosine	Alzheimer’s disease	NR	560 ± 203	NR	NR	NR	NR	32.4 ± 5	NR	[114]
Niosomes	Cholesterol, polyoxyethylene monostearate, and borneol	Ginkgolide B and puerarin	Alzheimer’s disease and Parkinson’s disease	Intravenous	142.65	0.261	NR	49.90	NR	NR	Higher C_max_, AUC, t_1/2,_ and MRT values (rats).	[115]
Niosomes	Buspirone	Cholesterol, Span 60, Carbopol 934P, hydroxypropyl methylcellulose K4M, and benzalkonium chloride	Anxiety disorders	Intranasal	181.9 ± 0.36	NR	−15.4	87.7 ± 0.66	NR	2600 ± 0.48 to 7800 ± 0.56	NR	[116]

Abbreviations: AUC—area under the curve; C_max_—maximum drug concentration; EE—encapsulation efficiency; MRT—mean residence time; NR—not reported; PD—pharmacodynamic; PDI—polydispersity index; PK—pharmacokinetics; t_1/2—_elimination half-life; T_max_—time to achieve maximum drug concentration; and ZP—zeta potential.

## Data Availability

Not applicable.

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
