# Peer review of "Liposome-Derived Nanosystems for the Treatment of Behavioral and Neurodegenerative Diseases: The Promise of Niosomes, Transfersomes, and Ethosomes for Increased Brain Drug Bioavailability"

_pharmaceuticals, 2023, doi:10.3390/ph16101424_

Round 1

Reviewer 1 Report

Summary

This review article delves into the utilization of distinct liposome-derived vesicular delivery systems, specifically niosomes, transfersomes, and ethosomes, in the context of addressing psychiatric and neurodegenerative disorders. The initial section provides an exploration of various neurological conditions, including Alzheimer's and Parkinson's diseases, anxiety disorders, depression, and schizophrenia. The subsequent section offers an introduction to these novel vesicular structures themselves, followed by the third section, which elucidates their practical application in the targeted delivery of pharmaceutical agents designed to treat the aforementioned neurological disorders discussed in the first section.

General comments

The document exhibits commendable writing quality and is relevant within its designated domain. While the cited articles are predominantly recent, many of them represent other review articles. Nonetheless, there exist several critical aspects requiring attention:

·         Uniqueness and Originality: A major concern is the abundance of existing reviews that center on various brain disorders and their association with liposomes and their derivatives. The authors need to accentuate the originality of their work, particularly in Section 4. Encouragingly, the authors should consider incorporating their own data, potentially elucidating novel and innovative approaches and highlight their expertise.

·         Improved Citation Placement: Throughout the document, numerous sections witness references being cited solely at the end of paragraphs. Authors should cite each reference within the text, contextualizing the source of information more effectively for the reader.

·         Organizational Enhancement: To enhance the document's organization, the inclusion of recapitulative tables containing the mode of administration, fabrication methods, targets, clinical outcomes and reference would be preferable over introducing figures from previously published papers that readers can access in the original articles. 

·         Route of Administration: The review should delve into discussions regarding the diverse routes of administration, such as intravenous, local, and intranasal delivery, considering the distinct challenges associated with each approach. Presently, the document amalgamates these aspects, rendering it challenging for readers to discern the nuances.

·         In-Depth Description of Brain Disorders: In Section 1, the authors briefly outline various brain disorders. However, given the complexity of these conditions, a more comprehensive elucidation is warranted. This section should encompass in-depth descriptions of the pathophysiological symptoms, underlying mechanisms, current clinical targets, and limitations of drug delivery systems.

·         Elaboration on Novel Approaches: In Section 2, where niosomes, ethosomes, and transferosomes are introduced as innovative solutions to overcome the limitations of conventional liposomes, the authors should provide more specific examples that underscore the enhanced effects of these vesicles, elaborate on their fabrication methods, and elucidate variations concerning their intended targets.

Specific comments:

·         Figure 1: The title should be amended to "Current Treatment Limitations" rather than "Current Treatments" to eliminate potential confusion. Additionally, while the figure is visually appealing, it presents a challenging amount of information to comprehend. If pathophysiological elements or symptoms are depicted, they should be appropriately described or labeled within the figure.

·         Section 1 (Lines 126-127): The statement, "It is a genetically inherited disease, and has most incidence in middle to late-life," should be clarified to specify which form of Alzheimer's disease is under consideration. Alzheimer's disease comprises two main forms: the hereditary form (typically linked to presenilin gene mutations), which represents a small fraction of cases, and the sporadic form which is believed to be associated with different isoforms of ApoE proteins. It's important to specify the form being discussed.

·         Section 1 (Lines 129-131): The passage discussing currently approved drugs for Alzheimer's disease should be updated to include emerging treatments, such as antibody-based therapies and ongoing research related to Tau-phosphorylation inhibitors, some of which involve drug delivery systems and are presently undergoing clinical trials. This will provide a more comprehensive view of the current Alzheimer's disease treatment.

·         Section 1 (Lines 141-143): The statement "The most commonly used pharmacological therapy is levodopa, but non-pharmacological treatments are also required in order to slow down the disease’s progression" should be expanded upon. It would be beneficial to provide specific examples of non-pharmacological treatments that are recommended for Parkinson's disease patients to slow the progression of the disease.

·         Section 3 (Line 185): When referring to "specific moieties," further elaboration and examples should be provided to enhance the reader's understanding of the term and its relevance in the context of the discussion.

·        Section 2, Figure 2: The figure should also include information about the mode of administration seen in the scientific litterature so far (e.g., oral, systemic, intranasal) for different types of nanosystems, providing a more comprehensive view of their practical applications. Additionally, while the figure is visually appealing, the authors should elaborate on the "features" provided in the upper-right corner.

·         Section 3.2 (Lines 350-352): The reference to insulin leading to Tau hyperphosphorylation, a symptom of Alzheimer's disease, should be addressed in more detail. Tau hyperphosphorylation is responsible for neurofibrillary entanglements leading to neuronal cell death. Insuline therefore appears to have a bad impact on Alzheimer's disease. Please elaborate.

·         Section 3.3 (Line 539): When referring to the Korsmeyer-Peppas model and its application to release kinetics, it is essential to elaborate on the various drug release mechanisms that this model can represent, depending on the value of the "n" exponential parameter. Additionally, specify the dominant mass transport phenomenon observed in the cited article to provide a clearer understanding in the context of the study.

Minor comments:

·       Keywords: "Alzheimer's disease" instead of Alzheimer’s

·       Section 4 (line 745), “As it is made”

N/A.

Author Response

We thank the reviewer for all their useful comments, suggestions and feedback, which have enabled us to improve our manuscript. A point-by-point answer is given bellow, and all changes in the manuscript have been highlighted in blue.

Summary

This review article delves into the utilization of distinct liposome-derived vesicular delivery systems, specifically niosomes, transfersomes, and ethosomes, in the context of addressing psychiatric and neurodegenerative disorders. The initial section provides an exploration of various neurological conditions, including Alzheimer's and Parkinson's diseases, anxiety disorders, depression, and schizophrenia. The subsequent section offers an introduction to these novel vesicular structures themselves, followed by the third section, which elucidates their practical application in the targeted delivery of pharmaceutical agents designed to treat the aforementioned neurological disorders discussed in the first section.

General comments

The document exhibits commendable writing quality and is relevant within its designated domain. While the cited articles are predominantly recent, many of them represent other review articles. Nonetheless, there exist several critical aspects requiring attention:

Uniqueness and Originality: A major concern is the abundance of existing reviews that center on various brain disorders and their association with liposomes and their derivatives. The authors need to accentuate the originality of their work, particularly in Section 4. Encouragingly, the authors should consider incorporating their own data, potentially elucidating novel and innovative approaches and highlight their expertise.

Answer: We thank the reviewer for their comment. The novelty of this manuscript resides in the fact that, to the best of our knowledge, no literature reviews have been made so far on the relevance of niosomes, transfersomes and ethosomes for the treatment of psychiatric and neurodegenerative disorders, more specifically highly impactful diseases such as schizophrenia, bipolar disorder, depression, anxiety, Alzheimer’s disease and Parkinson’s disease, although many reviews revolving around topical drug delivery, namely for the treatment of infections, skin cancer, or even for cosmetic purposes, already exist, as stated from lines 250 to 256. We have now further commented and highlighted this fact in section 4, as requested.

Improved Citation Placement: Throughout the document, numerous sections witness references being cited solely at the end of paragraphs. Authors should cite each reference within the text, contextualizing the source of information more effectively for the reader.

Answer: We thank the reviewer for their correction. Citation placement has now been improved accordingly.

Organizational Enhancement: To enhance the document's organization, the inclusion of recapitulative tables containing the mode of administration, fabrication methods, targets, clinical outcomes and reference would be preferable over introducing figures from previously published papers that readers can access in the original articles.

Answer: We thank the reviewer for their suggestion. We have now added a Table with such information, at the end of section 2, including nanosystem type, main formulation composition, encapsulated molecule, disease intended to treat, intended administration route, particle size, polydispersity index, zeta potential, encapsulation efficiency, pH, viscosity, main in vivo pharmacokinetic and/or pharmacodynamic results, and respective reference.

Route of Administration: The review should delve into discussions regarding the diverse routes of administration, such as intravenous, local, and intranasal delivery, considering the distinct challenges associated with each approach. Presently, the document amalgamates these aspects, rendering it challenging for readers to discern the nuances.

Answer: We thank the reviewer for their comment. Detailed information has now been added regarding each mentioned administration route, from lines 393 to 404, from lines 439 to 451, from lines 677 to 687, and from lines 861 to 870.

In-Depth Description of Brain Disorders: In Section 1, the authors briefly outline various brain disorders. However, given the complexity of these conditions, a more comprehensive elucidation is warranted. This section should encompass in-depth descriptions of the pathophysiological symptoms, underlying mechanisms, current clinical targets, and limitations of drug delivery systems.

Answer: We thank the reviewer for their recommendation. Detailed information has now been added accordingly, all through pages 2 to 5.

Elaboration on Novel Approaches: In Section 2, where niosomes, ethosomes, and transferosomes are introduced as innovative solutions to overcome the limitations of conventional liposomes, the authors should provide more specific examples that underscore the enhanced effects of these vesicles, elaborate on their fabrication methods, and elucidate variations concerning their intended targets.

Answer: We thank the reviewer for their suggestion. Additional relevant detailed information has now been added accordingly, on pages 7 and 8.

Specific comments:

Figure 1: The title should be amended to "Current Treatment Limitations" rather than "Current Treatments" to eliminate potential confusion. Additionally, while the figure is visually appealing, it presents a challenging amount of information to comprehend. If pathophysiological elements or symptoms are depicted, they should be appropriately described or labeled within the figure.

Answer: We thank the reviewer for their suggestion. The appropriate correction has been made, and the image has been replaced. As for depicted elements, the purpose for their inclusion was to simply place a relevant element for each disease, with them being reasonably self-explicative (neurodegeneration, mood swings, etc.). The specification of such elements is then done throughout the main body of the text.

Section 1 (Lines 126-127): The statement, "It is a genetically inherited disease, and has most incidence in middle to late-life," should be clarified to specify which form of Alzheimer's disease is under consideration. Alzheimer's disease comprises two main forms: the hereditary form (typically linked to presenilin gene mutations), which represents a small fraction of cases, and the sporadic form which is believed to be associated with different isoforms of ApoE proteins. It's important to specify the form being discussed.

Answer: We thank the reviewer for their comment. This information has now been corrected and completed, from lines 169 to 174.

Section 1 (Lines 129-131): The passage discussing currently approved drugs for Alzheimer's disease should be updated to include emerging treatments, such as antibody-based therapies and ongoing research related to Tau-phosphorylation inhibitors, some of which involve drug delivery systems and are presently undergoing clinical trials. This will provide a more comprehensive view of the current Alzheimer's disease treatment.

Answer: We thank the reviewer for their suggestion. This information has now been improved and completed, on lines 180 to 185, along with the respective references.

Section 1 (Lines 141-143): The statement "The most commonly used pharmacological therapy is levodopa, but non-pharmacological treatments are also required in order to slow down the disease’s progression" should be expanded upon. It would be beneficial to provide specific examples of non-pharmacological treatments that are recommended for Parkinson's disease patients to slow the progression of the disease.

Answer: We thank the reviewer for their comment. This information has now been added, on lines 205 to 207, along with the respective references.

Section 3 (Line 185): When referring to "specific moieties," further elaboration and examples should be provided to enhance the reader's understanding of the term and its relevance in the context of the discussion.

Answer: We thank the reviewer for their suggestion. A further elaboration on the matter has been now added, from lines 251 to 256, with the respective references also having been added.

Section 2, Figure 2: The figure should also include information about the mode of administration seen in the scientific litterature so far (e.g., oral, systemic, intranasal) for different types of nanosystems, providing a more comprehensive view of their practical applications. Additionally, while the figure is visually appealing, the authors should elaborate on the "features" provided in the upper-right corner.

Answer: We thank the reviewer for their suggestion. Figure 2 has now been changed to include the administration routes of the developed nanosystems. As for the features provided in the upper-right corner, they are elaborated, explained and specified in the text, all through section 2, since we feel that adding more text to the figure could make it less concise and more confusing for reader interpretation.

Section 3.2 (Lines 350-352): The reference to insulin leading to Tau hyperphosphorylation, a symptom of Alzheimer's disease, should be addressed in more detail. Tau hyperphosphorylation is responsible for neurofibrillary entanglements leading to neuronal cell death. Insuline therefore appears to have a bad impact on Alzheimer's disease. Please elaborate.

Answer: We thank the reviewer for the correction. It was in fact an incorrect information, since it is the depletion of insulin pathways that will lead to Tau hyperphosphorylation (and not the opposite), and hence the sentence has been rewritten (lines 494 to 497).

Section 3.3 (Line 539): When referring to the Korsmeyer-Peppas model and its application to release kinetics, it is essential to elaborate on the various drug release mechanisms that this model can represent, depending on the value of the "n" exponential parameter. Additionally, specify the dominant mass transport phenomenon observed in the cited article to provide a clearer understanding in the context of the study.

Answer: We thank the reviewer for their comment. A better elucidation of the mentioned in vitro release kinetic model would in fact be potentially useful, nevertheless, since no other article mentioned this topic, for the sake of uniformity, and also due to the specification of such models not being the purpose of this review, this information has been withdrawn. Additionally, the information regarding the in vitro drug release provided for this article has now been completed with data and wording that are more in accordance with the rest of the articles (cumulative drug release, and whether the release profile had controlled release characteristics or not). Changes are present from lines 692 to 694.

Minor comments:

Keywords: "Alzheimer's disease" instead of Alzheimer’s

Answer: We thank the reviewer for their suggestion, we have made the correction accordingly.

Section 4 (line 745), “As it is made”

Answer: We thank the reviewer for their correction, the change has been made in the text.

Reviewer 2 Report

A very interesting work, but contains a number of shortcomings.

1. Nanosystems are described as particles with size ranging from 1 to 100 nm, or particles that have one dimension below 10 nm (e.g. Duncan and Gaspar, 2011; Petros and DeSimone, 2010). In your work, they are defined as “structures with sizes between 1 and 1000 nm,” and most of the particles described in the review are larger than 100 nm. Authors must confirm, using references, that the systems described in the review belong to nanosystems or not use the term nanosystems and nanoparticles in their work (and also change the title of the review).

2. For a better perception of the information contained in the review, it is necessary to provide a summary table in the text, which would indicate the particles described in the review, the drug they contain, the disease, a reference to the work, as well as a number of other parameters that allow one to evaluate the properties of such systems .

3. Since brain bioavailability and penetration through the BBB are one of the key elements of the review, in the section “Liposome-derived vesicles: the future for brain drug delivery” should be summarized the mechanisms of penetration of the described particles into the brain in more detail (may be possible to make a table) especially since they are described in the text of the review.

4. In Figure 2, the orange molecule is labeled as cholesterol. This is right? Perhaps “lipid” would be more correct?

5. In section 3.1 “Ethosomes” only one substance is described - Ligustrazine. Perhaps a few more examples should be given.

Author Response

We thank the reviewer for all their useful comments, suggestions and feedback, which have enabled us to improve our manuscript. A point-by-point answer is given bellow, and all changes in the manuscript have been highlighted in blue.

A very interesting work, but contains a number of shortcomings.

  1. Nanosystems are described as particles with size ranging from 1 to 100 nm, or particles that have one dimension below 10 nm (e.g. Duncan and Gaspar, 2011; Petros and DeSimone, 2010). In your work, they are defined as “structures with sizes between 1 and 1000 nm,” and most of the particles described in the review are larger than 100 nm. Authors must confirm, using references, that the systems described in the review belong to nanosystems or not use the term nanosystems and nanoparticles in their work (and also change the title of the review).

Answer: We thank the reviewer for their comment. In fact, although the traditional convention is to classify nanoparticles as being between 1 to 100 nm, in more recent years many researchers have extended this range to 1 to 1000 nm. Nevertheless, we did not in fact have adequate references supporting this information. They have now been added, on line 245.

  1. For a better perception of the information contained in the review, it is necessary to provide a summary table in the text, which would indicate the particles described in the review, the drug they contain, the disease, a reference to the work, as well as a number of other parameters that allow one to evaluate the properties of such systems.

Answer: We thank the reviewer for their useful suggestion. We have now added a Table with such information, at the end of section 2, including nanosystem type, main formulation composition, encapsulated molecule, disease intended to treat, intended administration route, particle size, polydispersity index, zeta potential, encapsulation efficiency, pH, viscosity, main in vivo pharmacokinetic and/or pharmacodynamic results, and respective reference.

  1. Since brain bioavailability and penetration through the BBB are one of the key elements of the review, in the section “Liposome-derived vesicles: the future for brain drug delivery” should be summarized the mechanisms of penetration of the described particles into the brain in more detail (may be possible to make a table) especially since they are described in the text of the review.

Answer: We thank the reviewer for their suggestion. The requested information has now been added in the mentioned section.

  1. In Figure 2, the orange molecule is labeled as cholesterol. This is right? Perhaps “lipid” would be more correct?

Answer: We thank the reviewer for their correction. In fact, although cholesterol is one of the main lipids used for this effect, it is not the only one. Hence, Figure 2 has been corrected and replaced.

  1. In section 3.1 “Ethosomes” only one substance is described - Ligustrazine. Perhaps a few more examples should be given.

Answer: We thank the reviewer for their suggestion. Although there are many studies regarding the development of ethosomes for other applications, especially drug delivery on or through the skin - acne, psoriasis and other skin infections, or wound healing (https://pubmed.ncbi.nlm.nih.gov/35090058/, https://pubmed.ncbi.nlm.nih.gov/22297749/, (https://pubmed.ncbi.nlm.nih.gov/17884226/, https://pubmed.ncbi.nlm.nih.gov/35631628/, https://pubmed.ncbi.nlm.nih.gov/34631355/), and even cosmetic purposes (https://pubmed.ncbi.nlm.nih.gov/37393573/) - there is a severe scarcity of studies developing ethosomes for the treatment of psychiatric and neurodegenerative diseases. This is the reason why we only added 1 study to the “Ethosomes” section, because it was the only one that we found that was suitable for the scope of this review. 

Round 2

Reviewer 1 Report

The revised version of the manuscript  has resolved the main issues that were pointed out in the previous one.

Reviewer 2 Report

Thanks for the detailed answer